# KG-Infused RAG: Augmenting Corpus-Based RAG with External Knowledge Graphs

## Abstract

Retrieval-Augmented Generation (RAG) improves factual accuracy by grounding responses in external knowledge. However, existing RAG methods either rely solely on text corpora and neglect structural knowledge, or build ad-hoc knowledge graphs (KGs) at high cost and low reliability. To address these issues, we propose **KG-Infused RAG**, a framework that incorporates pre-existing large-scale KGs into RAG and applies *spreading activation* to enhance both retrieval and generation. KG-Infused RAG directly performs spreading activation over external KGs to retrieve relevant structured knowledge, which is then used to expand queries and integrated with corpus passages, enabling interpretable and semantically grounded multi-source retrieval. We further improve KG-Infused RAG through preference learning on sampled key stages of the pipeline. Experiments on five QA benchmarks show that KG-Infused RAG consistently outperforms vanilla RAG (by 3.9% to 17.8%). Compared with KG-based approaches such as GraphRAG and LightRAG, our method obtains structured knowledge at lower cost while achieving superior performance. Additionally, integrating KG-Infused RAG with Self-RAG and DeepNote yields further gains, demonstrating its effectiveness and versatility as a plug-and-play enhancement module for corpus-based RAG methods.

## 1 Introduction

Large language models (LLMs) have shown strong performance in question answering tasks (Hurst et al., 2024; Grattafiori et al., 2024; Yang et al., 2024), but remain prone to factual errors and hallucinations (Mallen et al., 2023; Min et al., 2023; Gao et al., 2023b). Retrieval-Augmented Generation (RAG) mitigates these issues by grounding generation in external knowledge sources (Lewis et al., 2020; Trivedi et al., 2023; Asai et al., 2024; Wen et al., 2024). Existing RAG methods can be divided into two main categories: (1) corpus-based approaches that retrieve passages from large-scale text corpora (Gao et al., 2023b; Trivedi et al., 2023; Asai et al., 2024), providing broad knowledge coverage but neglecting important structural information, and (2) KG-based methods (Edge et al., 2024; Guo et al., 2024) that use LLMs to construct ad-hoc knowledge graphs (KGs) from the corpus to enhance the retrieval process, but are limited by retrieval efficiency and the limited amount of information in the constructed KGs. However, both approaches fail to leverage existing large-scale curated KGs, which could enhance both the coverage and efficiency of retrieval.

To bridge these gaps, we propose **Knowledge Graph Infused Retrieval-Augmented Generation (KG-Infused RAG)**, a framework that leverages pre-existing, large-scale KGs to enhance retrieval and generation. At its core, KG-Infused RAG employs *spreading activation* (Collins & Loftus, 1975), a concept from cognitive psychology in which activation propagates from a central concept to related ones in a semantic network. In RAG, this idea allows LLMs to simulate spreading activation by retrieving structured knowledge from KGs, activating key facts, and progressively accessing query-relevant information. By exploiting KG structure, KG-Infused RAG links related entities and facts often missed by corpus-based methods, leading to more accurate and context-aware retrieval.

Furthermore, KG-Infused RAG eliminates the need for ad-hoc KG construction via LLMs (Edge et al., 2024; Guo et al., 2024; Zhu et al., 2025), which are computationally expensive and prone to factual inaccuracies. Instead, KG-Infused RAG directly utilizes pre-existing KGs, ensuring efficient and reliable retrieval. This use of large-scale KGs enhances retrieval coverage and interpretability, providing a more structured and accurate foundation for downstream generation.

KG-Infused RAG consists of three stages: (1) retrieving relevant KG facts via spreading activation, (2) expanding the query with these facts to improve corpus retrieval, (3) generating answers with passages enriched by KG facts. This multi-source retrieval combines KG facts and corpus passages, leveraging KG structure to guide spreading activation. By aligning the retrieval process with spreading activation, KG-Infused RAG enables more accurate and fact-grounded QA. Compared with KG-based methods such as GraphRAG and LightRAG, KG-Infused RAG avoids costly LLM-based KG construction, achieves more efficient inference, and delivers stronger performance. Additionally, it can be integrated into corpus-based RAG methods like Self-RAG (Asai et al., 2024) to further boost performance.

To further improve KG-Infused RAG, we apply preference learning on sampled key pipeline stages for targeted tuning. Experiments on multiple benchmarks demonstrate consistent performance gains and show that KG-Infused RAG significantly enhances retrieval quality compared to vanilla RAG.

Our main contributions are summarized as follows:

1. We introduce KG-Infused RAG, a framework that incorporates pre-existing KGs into RAG to apply spreading activation, enabling LLMs to retrieve relevant structured knowledge from KGs.
2. Experiments on five QA benchmarks show that KG-Infused RAG consistently improves performance, with absolute gains of 3.9% to 17.8% over vanilla RAG.
3. KG-Infused RAG outperforms GraphRAG and LightRAG in both efficiency and performance.
4. Plug-in application: KG-Infused RAG can be integrated into corpus-based RAG methods (e.g., Self-RAG and DeepNote) as a plug-and-play module.

## 2 RELATED WORK

### 2.1 MULTI-TURN RETRIEVAL IN RETRIEVAL-AUGMENTED GENERATION

Single-turn retrieval followed by generation often yields insufficient evidence for complex questions, such as multi-hop QA, which requires multiple retrieval steps and the integration and reasoning of retrieved information (Gao et al., 2023b; Trivedi et al., 2023).

**Query Rewriting and Expansion.** Query rewriting reformulates ambiguous or complex queries into clearer, more effective ones (Ma et al., 2023; Mao et al., 2024). Self-Ask (Press et al., 2023) and IRCoT (Trivedi et al., 2023) enhance multi-hop reasoning by decomposing complex queries into sub-questions for iterative retrieval. Query expansion, such as adding keywords or pseudo-documents, enriches the original query to improve retrieval (Wang et al., 2023; Jagerman et al., 2023; Lei et al., 2024). HyDE (Gao et al., 2023a) generates hypothetical documents from the query to improve retrieval relevance. Unlike these methods that rely solely on an LLM's parametric knowledge, KG-Infused RAG incorporates factual signals from a knowledge graph (KG) to ground query expansion, KG-Infused RAGenables more targeted and evidence-grounded retrieval.

**Evidence Aggregation and Enhancement.** Recent RAG methods extend single-turn retrieval to multi-turn, collecting richer and more targeted evidence (Gao et al., 2023b). IRCoT interleaves retrieval with chain-of-thought reasoning steps, while FLARE (Jiang et al., 2023) and Self-RAG (Asai et al., 2024) dynamically decide when and what to retrieve based on generation confidence or intermediate outputs. These methods enable incremental evidence accumulation. Similarly, KG-Infused RAG gathers additional evidence through multi-turn spreading activation and KG-based query expansion, followed by integration and refinement of retrieved information.

### 2.2 KNOWLEDGE GRAPHS AUGMENTED QUESTION ANSWERING

Knowledge graphs (KGs) provide structured representations of entities and their relationships, which help reduce hallucinations, improve reasoning interpretability, and enhance retrieval and generation in RAG systems (Baek et al., 2023; Xu et al., 2024; Wen et al., 2024; Guo et al., 2024). MindMap (Wen et al., 2024) and Xu et al. (2024) use KGs as retrieval sources in domains like medical and customer service QA, achieving better retrieval accuracy, answer quality, and transparency. In contrast, GraphRAG (Edge et al., 2024), LightRAG (Guo et al., 2024), and KG²RAG (Zhu et al., 2025) generate ad-hoc KGs from the corpus or retrieved passages to organize entity relationships and improve answer coherence for complex questions. However, this requires significant computational

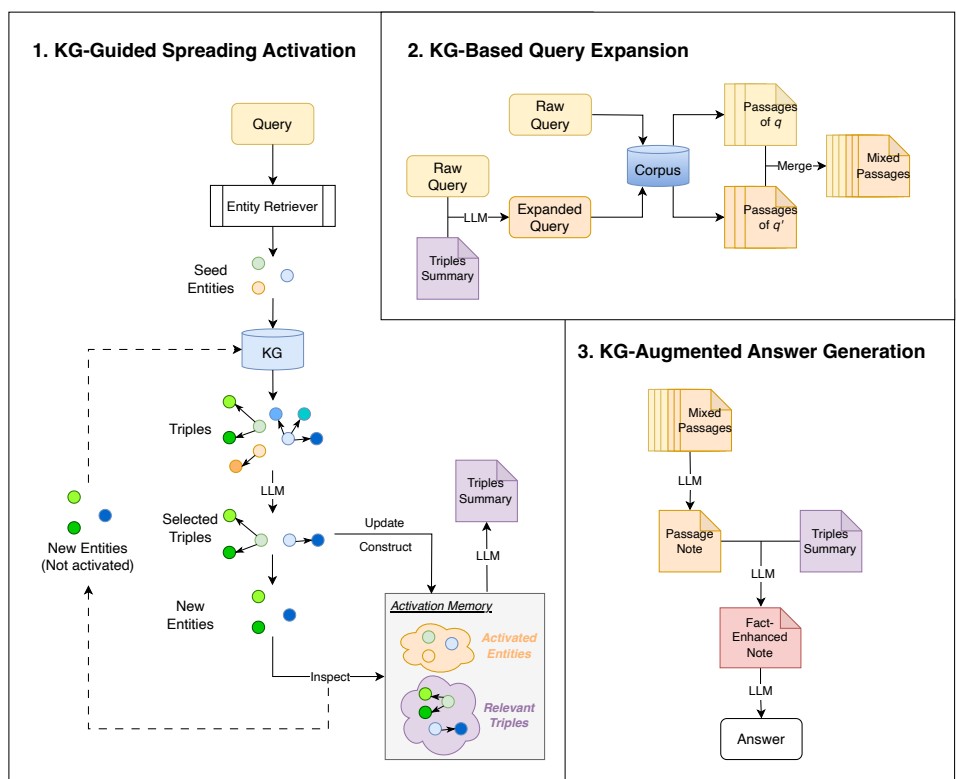

Figure 1: **Overview of KG-Infused RAG.** The framework consists of three modules: (1) KG-Guided Spreading Activation, (2) KG-Based Query Expansion, and (3) KG-Augmented Answer Generation, which together enable interpretable, fact-grounded multi-source retrieval.

resources for KG construction, making it impractical for large-scale corpora. KG-Infused RAG, on the other hand, directly use pre-existing, large-scale KGs, avoiding the costly construction process while providing reliable, structured, and board external knowledge.

## 3 METHODOLOGY

### 3.1 TASK DEFINITION

Given a question $q$, a KG $\mathcal{G}$, and a corpus $\mathcal{D}_c$, the goal is to generate an answer $a$ by retrieving structured facts from $\mathcal{G}$ and textual passages from $\mathcal{D}_c$, and integrating them for final answer generation.

Let $\mathcal{G}_q \subset \mathcal{G}$ denote the retrieved KG subgraph, consisting of query-relevant triples, and $\mathcal{P}_q \subset \mathcal{D}_c$ represent the corresponding passages that are retrieved. Considering the KG as a semantic network of concept-level associations, the entities within $\mathcal{G}_q$ act as seeds, facilitating the propagation of activation throughout the KG. Formally, the task can be framed as:

$$\text{answer} \sim \text{LLM}\left(q, \mathcal{G}_q, \mathcal{P}_q\right) \tag{1}$$

### 3.2 FRAMEWORK OVERVIEW

We propose KG-Infused RAG, a framework inspired by the cognitive process of spreading activation, where activation spreads from a central concept to semantically related ones. By leveraging the rich semantic structure of pre-existing, large-scale KGs, KG-Infused RAG extends relevant knowledge via multi-hop associations, enhancing the retrieval and integration of both KG facts and textual passages, without the need for ad-hoc KG construction. The overall process is illustrated in Figure 1, consisting of three main modules:

1) **KG-Guided Spreading Activation.** Starting from query-relevant entities, we iteratively expand related triples to form a task-specific subgraph $\mathcal{G}_q$, which is summarized for downstream use.

2) **KG-Based Query Expansion (KG-Based QE).** The LLM leverages both the original query and the expanded KG subgraph to generate an expanded query, improving retrieval over the corpus $\mathcal{D}_c$.

3) **KG-Augmented Answer Generation (KG-Aug Gen).** The LLM generates the answer by integrating retrieved passages and KG summaries, ensuring fact-grounded and interpretable generation.

We prepare our KG $\mathcal{G}$ by extracting and preprocessing triples from a large-scale knowledge graph dataset, Wikidata5M. Details of the preprocessing are provided in Appendix C. Details of all prompts used in the framework are in Appendix E.1.

### 3.3 KG-GUIDED SPREADING ACTIVATION

This stage constructs a query-specific subgraph $\mathcal{G}_q$ by simulating spreading activation over the KG, starting from query-relevant entities and propagating through related facts. The retrieved structured knowledge complements corpus evidence and supports downstream retrieval and generation.

**Seed Entities Initialization.** To initialize the spreading activation process, we first retrieve the top-$k_e$ entities most relevant to the query $q$. Each entity $e_i$ is associated with a textual description $d_i \in \mathcal{D}_e$, and we compute the similarity between the query and each description using inner product:

$$\text{sim}(q, d_i) = q \cdot d_i \tag{2}$$

The top-$k_e$ entities with the highest similarity scores are selected:

$$E_q^0 = \{e_i \mid d_i \in \text{TopK}_{k_e}(\mathcal{D}_e; \text{sim}(q, d_i))\} \tag{3}$$

where $\text{TopK}_{k_e}(\mathcal{D}_e; \text{sim})$ denotes the set of $k_e$ descriptions in $\mathcal{D}_e$ with the highest similarity to $q$. The resulting entity set $E_q^0 = \{e_1, e_2, \ldots, e_{k_e}\}$ initializes the spreading activation process over the KG.

**Iterative Spreading Activation.** The spreading activation process begins with the seed entities $E_q^0$ and proceeds iteratively over $\mathcal{G}$. In each round $i$, activation expands from the currently activated entities $E_q^i$, retrieving additional query-relevant triples. Each round consists of the following steps:

1. **Triple Selection.** At each round, for every entity in the current set $E_q^i$, we retrieve its 1-hop neighbors from $\mathcal{G}$. An LLM is then prompted to select a subset of triples relevant to the query $q$. The selected triples for round $i$ are denoted as $\{(e, r, e')_i\}$, where $e$, $r$, and $e'$ represent the head entity, relation, and tail entity in a triple, respectively.

2. **Activation Memory Construction and Updating.** We maintain an **Activation Memory** $\mathcal{M} = \{\mathcal{G}_q^{act}, E_q^{act}\}$, which tracks the multi-round activation process. The subscript $act$ denotes the activation stage. Here, $\mathcal{G}_q^{act}$ is the accumulated subgraph of all query-relevant triples retrieved up to the current round, and $E_q^{act}$ is the set of entities activated thus far. This memory serves to aggregate knowledge and prevent redundant reactivation of entities. At each round $i$, given the newly retrieved triples $\mathcal{G}_q^i$ and the associated entities $E_q^i$, we update the memory as follows:

$$\mathcal{G}_q^{act} \leftarrow \mathcal{G}_q^{act} \cup \mathcal{G}_q^i \quad \text{and} \quad E_q^{act} \leftarrow E_q^{act} \cup E_q^i \tag{4}$$

where $\cup$ denote the set union operation.

3. **Next-Round Activation Entities.** At each round $i$, we determine the next-round entity set $E_q^{i+1}$ from the current triples $\mathcal{G}_q^i = \{(e, r, e')_i\}$ by extracting their tail entities $\{e'\}$ as candidates for further activation. To prevent revisiting entities, we exclude those already present in the activated entity set $E_q^{act}$. Formally,

$$E_q^{i+1} = \left( \bigcup_{(e, r, e') \in \mathcal{G}_q^i} \{e'\} \right) \setminus E_q^{act} \tag{5}$$

where the set difference $\setminus$ excludes previously activated entities, ensuring $E_q^{i+1}$ contains only newly activated entities to guide the next activation round.

The activation process terminates when either the predefined maximum number of activation rounds $k$ is reached or no new entities remain, i.e., when $E_q^{i+1} = \emptyset$. This iterative process progressively retrieves query-relevant triples from $\mathcal{G}$, spanning paths from 1-hop to $k$-hop neighborhoods.

**Expanded Subgraph Summarization.** We prompt the LLM to summarize the expanded KG subgraph $\mathcal{G}_q^{act}$, accumulated in the Activation Memory $\mathcal{M}$, into a natural language summary $\mathcal{S}_{\mathcal{G},q}^{act}$. This summarization serves two purposes: **(1)** It condenses discrete graph-structured facts into a coherent natural language narrative, uncovering semantic paths and concept interactions underlying the query. **(2)** It converts structured knowledge into natural language, making it more accessible and usable for the LLM in the subsequent stages.

## 3.4 KG-BASED QUERY EXPANSION

The goal of this stage is to generate an expanded query $q'$ which complements the original query $q$ by incorporating structured knowledge retrieved from the KG. The expansion broadens the retrieval coverage and enhances the relevance of corpus evidence.

We prompt the LLM with both the original query $q$ and the KG subgraph summary $\mathcal{S}_{\mathcal{G},q}^{act}$ constructed via spreading activation. Conditioned on the prompt, the model performs associative reasoning to transform the query—either simplifying it based on retrieved facts (e.g., replacing intermediate entities), or extending it with new, KG-grounded content inferred from LLM knowledge. Formally,

$$q' \sim \text{LLM}\left(\text{Instruct}_{\text{QE}}, q \parallel \mathcal{S}_{\mathcal{G},q}^{act}\right) \tag{6}$$

Where $\text{Instruct}_{\text{QE}}$ is the prompt template used for query expansion. We then perform dual-query retrieval over the corpus using both $q$ and $q'$, and merge the results:

$$\mathcal{D}_{c,(q,q')} = \mathcal{D}_{c,q} \cup \mathcal{D}_{c,q'} \tag{7}$$

where $\mathcal{D}_{c,q}$ and $\mathcal{D}_{c,q'}$ denote the passage sets retrieved by $q$ and $q'$, respectively.

## 3.5 KG-AUGMENTED ANSWER GENERATION

This stage integrates both corpus and KG evidence to generate the final answer. By augmenting the retrieved passages with structured facts, we facilitate more informed and accurate reasoning.

**Passage Note Construction.** We first prompt the LLM to summarize the retrieved passages $\mathcal{D}_{c,(q,q')}$, generating a query-focused passage note $\mathcal{S}_{\mathcal{D},q}$ that distills key information relevant to answering $q$:

$$\mathcal{S}_{\mathcal{D},q} \sim \text{LLM}\left(\text{Instruct}_{\text{Note}}, q \parallel \mathcal{D}_{c,(q,q')}\right) \tag{8}$$

where $\text{Instruct}_{\text{Note}}$ is the prompt template used for passage note construction.

**KG-Guided Knowledge Augmentation (KG-Guided KA).** To incorporate structured knowledge, we prompt the LLM to augment the passage note $\mathcal{S}_{\mathcal{D},q}$ with the KG subgraph summary $\mathcal{S}_{\mathcal{G},q}^{act}$, yielding a fact-enhanced note $\mathcal{S}_q^{\text{final}}$:

$$\mathcal{S}_q^{\text{final}} \sim \text{LLM}\left(\text{Instruct}_{\text{KA}}, \mathcal{S}_{\mathcal{D},q} \parallel \mathcal{S}_{\mathcal{G},q}^{act}\right) \tag{9}$$

where $\text{Instruct}_{\text{KA}}$ is the prompt template used for knowledge augmentation.

**Answer Generation.** Finally, the LLM generates the answer conditioned on the original query $q$ and the enriched note $\mathcal{S}_q^{\text{final}}$, as follows:

$$\text{answer} \sim \text{LLM}\left(\text{Instruct}_{\text{Ans}}, q \parallel \mathcal{S}_q^{\text{final}}\right) \tag{10}$$

where $\text{Instruct}_{\text{Ans}}$ is the prompt template used for answer generation. This design enables the model to generate answers grounded not only in textual evidence but also in structured KG facts, facilitating more comprehensive, accurate, and interpretable reasoning.

## 3.6 DIRECT PREFERENCE OPTIMIZATION FOR KNOWLEDGE AUGMENTATION

Inspired by previous works (Li et al., 2025; Wang et al., 2024), we enhance the instruction-following capability of the model at the KG-Guided KA stage of KG-Infused RAG's pipeline by using Direct Preference Optimization (DPO) (Rafailov et al., 2023), which aligns with better response preferences, thereby enabling better integration of knowledge retrieved from both the KG and the corpus.

We construct a simple, preference-based training dataset $D_{\text{KA}}$ by sampling the response from the KG-Guided KA stage. The training objective is defined as:

$$\mathcal{L}_{\text{DPO}} = -\mathbb{E}_{(x,y^+,y^-)\sim D}\left[\log\sigma\left(\beta\left(\log\frac{\pi_{\text{KI}}^{\theta}(y^+|x)}{\pi_{\text{KI}}^{\text{ref}}(y^+|x)} - \log\frac{\pi_{\text{KI}}^{\theta}(y^-|x)}{\pi_{\text{KI}}^{\text{ref}}(y^-|x)}\right)\right)\right] \tag{11}$$

where $\pi_{\text{KI}}^{\theta}$ denotes the trainable model, and $\pi_{\text{KI}}^{\text{ref}}$ is a fixed reference model during training. Training data construction and additional training details are provided in Appendix F.

## 4 Experiment Setup

### 4.1 Datasets & Evaluation

**Multi-Hop QA Task.** To evaluate our method in complex QA scenarios, we conduct experiments on four challenging multi-hop QA datasets: HotpotQA (Yang et al., 2018), 2WikiMultihopQA (2WikiMQA) (Ho et al., 2020), MuSiQue (Trivedi et al., 2022), and Bamboogle (Press et al., 2023). For HotpotQA, 2WikiMQA, and MuSiQue, we use the subsets provided by Trivedi et al. (2023). These datasets require models to perform complex multi-hop reasoning across multiple pieces of evidence, making them ideal for evaluating our method's ability to integrate and reason over knowledge. We report F1 Score (F1), Exact Match (EM), Accuracy (Acc), and their average (Avg) as evaluation metrics. The summary of datasets and evaluation is in Appendix D.1.

**Commonsense QA Task.** To assess the generalizability of our method to simpler QA scenarios, we evaluate it on the StrategyQA dataset (Geva et al., 2021), a commonsense QA task requiring binary Yes/No answers. Unlike multi-hop QA tasks that demand explicit retrieval and integration of multiple evidence pieces, StrategyQA emphasizes implicit reasoning based on implicit commonsense knowledge. From the RAG perspective, it poses a lower retrieval challenge. We randomly sample 500 examples from the test set for evaluation, with accuracy as the evaluation metric.

### 4.2 Baselines

**Standard Baselines.** We compare our method against four types of baselines: (1) **No-Retrieval (NoR)**, which directly feeds the query into an LLM to generate the answer. (2) **Vanilla RAG**, which retrieves the top-$k_p$ passages from the corpus using the raw query $q$, and generate the answer. (3) **Vanilla Query Expansion (Vanilla QE)**, which first prompts an LLM to generate an expanded query $q'$ without leveraging KG facts, then both the raw and the expanded queries are used to retrieve top-$k_p/2$ passages each, and generate the answer. (4) **KG-Based methods**, which first use LLMs to construct an ad-hoc KG, then perform retrieval to generate the answer. For Vanilla RAG and Vanilla QE, we adopt our passage note construction step to ensure a fair comparison with KG-Infused RAG. For KG-Based methods, we select three representative baselines, including GraphRAG (Edge et al., 2024), LightRAG (Guo et al., 2024), and KG$^2$RAG (Zhu et al., 2025).

**Plug-in Baselines.** To evaluate the generalizability of our method as a plug-and-play module, we integrate it into three representative adaptive RAG methods: Self-RAG (Asai et al., 2024), Self-Ask (Press et al., 2023), and DeepNote (Wang et al., 2024). Specifically, we incorporate the KG-Guided Spreading Activation module into these baselines, enabling them to retrieve semantically related concepts from the KG, and further employ KG-Based QE and KG-Aug Gen to enhance their performance. Detailed integration procedures for each baseline are provided in Appendix G.

### 4.3 Resources and Model Components

**KG & Corpus.** For the KG $\mathcal{G}$ used in KG-Infused RAG, we adopt Wikidata5M-KG, an open-domain KG derived from Wikidata and Wikipedia. To ensure fair comparisons, we use the same corpus $\mathcal{D}_c$ within each group of baselines: (1) for non-KG-based baselines, we use the large-scale Wikipedia-2018 corpus containing 21,015,324 passages[1]; (2) for KG-based baselines, given the high cost of KG construction, we employ a smaller corpus by concatenating the HotpotQA, 2WikiMQA, and MuSiQue corpora provided in HippoRAG2 (Gutiérrez et al., 2024), resulting in 27,572 passages.

---

[1] https://github.com/facebookresearch/DPR

Table 1: **Overall results (%)** on five datasets. **Bold** numbers denote the best-performed results. We report KG-Infused RAG with two activation rounds ("w/ 2-Round"), where "w/ n-Round" denotes KG-Infused RAG equipped with $n$ rounds of activation.

| Method & LLMs | Multi-hop | | | | | | | | | | | | | | | | Commonsense |
| --- | --- | --- | --- | --- | --- | --- | --- | --- | --- | --- | --- | --- | --- | --- | --- | --- | --- |
| | HotpotQA | | | | 2WikiMQA | | | | MuSiQue | | | | Bamboogle | | | | StrategyQA |
| | Acc | F1 | EM | Avg | Acc | F1 | EM | Avg | Acc | F1 | EM | Avg | Acc | F1 | EM | Avg | Acc |
| *No-Retrieval (NoR)* | | | | | | | | | | | | | | | | | |
| Qwen2.5-3B-Instruct | 15.4 | 21.1 | 14.8 | 17.1 | 25.0 | 28.3 | 24.4 | 25.9 | 1.2 | 6.8 | 0.6 | 2.9 | 6.4 | 9.0 | 3.2 | 6.2 | 61.2 |
| Qwen2.5-7B-Instruct | 19.8 | 25.8 | 18.0 | 21.2 | 23.4 | 26.8 | 22.0 | 24.1 | 4.0 | 11.6 | 3.4 | 6.3 | 10.4 | 16.6 | 8.0 | 11.7 | 64.4 |
| LLaMA3.1-8B-Instruct | 22.2 | 27.0 | 20.6 | 23.3 | 28.0 | 29.3 | 24.4 | 27.2 | 3.6 | 8.8 | 3.0 | 5.1 | 11.2 | 16.4 | 8.0 | 11.9 | 68.4 |
| Qwen2.5-14B-Instruct | 22.2 | 28.8 | 21.8 | 24.3 | 28.6 | 31.2 | 27.6 | 29.1 | 5.6 | 12.0 | 4.4 | 7.3 | 13.6 | 20.8 | 12.0 | 15.5 | 64.4 |
| GPT4o-mini | 31.0 | 38.9 | 29.4 | 33.1 | 29.4 | 32.6 | 25.6 | 29.2 | 7.4 | 15.1 | 5.4 | 9.3 | 19.2 | 28.1 | 18.4 | 21.9 | 74.6 |
| LLaMA3.1-70B-Instruct | 31.8 | 40.5 | 30.4 | 34.2 | 32.6 | 36.5 | 30.4 | 33.2 | 8.0 | 13.8 | 6.4 | 9.4 | 32.0 | 41.5 | 28.8 | 34.1 | 72.0 |
| *Vanilla RAG* | | | | | | | | | | | | | | | | | |
| Qwen2.5-3B-Instruct | 28.2 | 34.8 | 26.0 | 29.7 | 29.0 | 32.3 | 27.2 | 29.5 | 3.8 | 9.6 | 3.2 | 5.5 | 12.0 | 19.6 | 9.6 | 13.7 | 62.6 |
| Qwen2.5-7B-Instruct | 35.0 | 41.9 | 31.0 | 36.0 | 30.6 | 33.5 | 27.2 | 30.4 | 5.2 | 11.5 | 3.6 | 6.8 | 22.4 | 30.1 | 21.6 | 24.7 | 68.6 |
| LLaMA3.1-8B-Instruct | 32.2 | 39.0 | 28.8 | 33.4 | 30.6 | 31.6 | 25.0 | 29.1 | 3.6 | 9.6 | 2.0 | 5.1 | 20.0 | 24.8 | 18.4 | 21.1 | 66.2 |
| Qwen2.5-14B-Instruct | 35.2 | 42.6 | 32.4 | 36.7 | 32.6 | 30.1 | 23.6 | 28.8 | 6.8 | 11.8 | 4.4 | 7.7 | 26.4 | 34.8 | 25.6 | 28.9 | 70.6 |
| GPT4o-mini | 43.0 | 52.4 | 39.6 | 45.0 | 33.6 | 36.0 | 28.4 | 32.7 | 9.8 | 18.1 | 8.0 | 12.0 | 32.0 | 41.1 | 30.4 | 34.5 | 74.4 |
| LLaMA3.1-70B-Instruct | 37.4 | 46.0 | 34.8 | 39.4 | 28.6 | 29.6 | 24.6 | 27.6 | 8.2 | 13.0 | 7.0 | 9.4 | 29.6 | 35.8 | 29.6 | 31.7 | 71.0 |
| *Vanilla Query Expansion (Vanilla QE)* | | | | | | | | | | | | | | | | | |
| Qwen2.5-3B-Instruct | 27.6 | 34.6 | 25.4 | 29.2 | 28.6 | 32.7 | 27.2 | 29.5 | 4.2 | 9.4 | 3.6 | 5.7 | 15.2 | 23.8 | 13.6 | 17.5 | 61.4 |
| Qwen2.5-7B-Instruct | 35.0 | 41.8 | 30.8 | 35.9 | 28.6 | 31.5 | 25.4 | 28.5 | 5.4 | 11.1 | 2.2 | 6.2 | 20.8 | 28.6 | 18.4 | 22.6 | 72.2 |
| LLaMA3.1-8B-Instruct | 32.6 | 38.8 | 28.8 | 33.4 | 26.6 | 28.6 | 21.6 | 25.6 | 5.2 | 11.0 | 3.4 | 6.5 | 18.4 | 23.3 | 15.2 | 19.0 | 68.0 |
| Qwen2.5-14B-Instruct | 35.0 | 42.4 | 32.0 | 36.5 | 32.6 | 31.1 | 25.2 | 29.6 | 7.6 | 13.3 | 5.0 | 8.6 | 29.6 | 35.4 | 29.6 | 31.5 | 72.4 |
| GPT4o-mini | 43.0 | 52.2 | 38.8 | 44.7 | 34.4 | 37.8 | 30.2 | 34.1 | 11.0 | 18.6 | 9.2 | 12.9 | 32.8 | 40.9 | 31.2 | 35.0 | 74.6 |
| LLaMA3.1-70B-Instruct | 38.4 | 45.9 | 36.0 | 40.1 | 26.8 | 28.0 | 23.4 | 26.1 | 11.2 | 15.2 | 8.6 | 11.7 | 33.6 | 39.4 | 31.2 | 34.7 | 72.6 |
| *KG-Infused RAG (ours)* | | | | | | | | | | | | | | | | | |
| KG-Infused RAG (Qwen2.5-3B-Instruct) | 28.0 | 34.6 | 25.4 | 29.3 | 39.0 | 41.8 | 34.0 | 38.3 | 8.6 | 14.0 | 7.0 | 9.9 | 16.0 | 23.9 | 14.4 | 18.1 | 61.4 |
| KG-Infused RAG (Qwen2.5-7B-Instruct) | 39.0 | 44.7 | 34.4 | 39.4 | 44.8 | 44.5 | 36.0 | 41.8 | 10.8 | 16.7 | 7.6 | 11.7 | 26.4 | 34.1 | 24.0 | 28.2 | 68.2 |
| +DPO | 37.2 | 43.9 | 31.0 | 37.4 | 45.6 | 45.8 | 35.8 | 42.4 | 12.9 | 19.2 | 8.6 | 13.5 | 36.8 | 41.8 | 31.2 | 36.6 | 72.8 |
| KG-Infused RAG (LLaMA3.1-8B-Instruct) | 34.4 | 39.8 | 30.2 | 34.8 | 37.0 | 36.0 | 27.4 | 33.5 | 12.0 | 16.9 | 7.4 | 12.1 | 24.0 | 28.5 | 22.4 | 25.0 | 67.0 |
| +DPO | 36.6 | 43.8 | 31.0 | 37.1 | 45.0 | 46.8 | 37.0 | 42.9 | 12.2 | 18.5 | 8.8 | 13.2 | 31.2 | 33.6 | 24.8 | 29.9 | 69.4 |
| $\Delta$ KG-Infused RAG+DPO→Vanilla RAG | 4.4↑ | 4.8↑ | 2.2↑ | 3.7↑ | 14.4↑ | 15.2↑ | 12.0↑ | 13.8↑ | 8.6↑ | 8.9↑ | 6.8↑ | 8.1↑ | 11.2↑ | 8.8↑ | 6.4↑ | 8.8↑ | 3.2↑ |
| KG-Infused RAG (Qwen2.5-14B-Instruct) | 38.0 | 45.6 | 35.4 | 39.7 | 47.4 | 44.9 | 35.6 | 42.6 | 14.0 | 19.0 | 9.4 | 14.1 | 31.2 | 42.6 | 29.6 | 34.5 | 75.2 |
| KG-Infused RAG (GPT4o-mini) | **44.0** | **53.5** | **39.8** | **45.8** | 47.6 | 47.5 | 38.0 | 44.4 | 15.0 | 21.9 | 10.8 | 15.9 | 38.4 | 43.3 | 36.0 | 39.2 | **75.4** |
| KG-Infused RAG (LLaMA3.1-70B-Instruct) | 41.6 | 50.1 | 38.2 | 43.3 | **49.0** | **49.2** | **40.8** | **46.3** | **17.0** | **22.7** | **13.0** | **17.6** | **48.8** | **54.2** | **45.6** | **49.5** | 75.0 |
| $\Delta$ KG-Infused RAG→Vanilla RAG | 4.2↑ | 4.1↑ | 3.4↑ | 3.9↑ | 20.4↑ | 19.6↑ | 16.2↑ | 18.7↑ | 8.8↑ | 9.7↑ | 6.0↑ | 8.2↑ | 19.2↑ | 18.4↑ | 16.0↑ | 17.8↑ | 4.0↑ |

Table 2: **Results (%) of KG-based methods.** "local" and "global" indicate different search modes used in these methods.

| Method | HotpotQA | | | | 2WikiMQA | | | | MuSiQue | | | | Bamboogle | | | |
| --- | --- | --- | --- | --- | --- | --- | --- | --- | --- | --- | --- | --- | --- | --- | --- | --- |
| | Acc | F1 | EM | Avg | Acc | F1 | EM | Avg | Acc | F1 | EM | Avg | Acc | F1 | EM | Avg |
| GraphRAG (local) (GPT-4o-mini) | 15.0 | 15.3 | 10.4 | 13.6 | 13.8 | 8.7 | 5.8 | 9.4 | 5.6 | 7.0 | 3.0 | 5.2 | 12.8 | 12.2 | 9.6 | 11.5 |
| GraphRAG (global) (GPT-4o-mini) | 32.4 | 28.9 | 20.2 | 27.2 | 31.2 | 19.2 | 13.2 | 21.2 | 10.8 | 9.4 | 3.0 | 7.7 | **44.0** | 34.0 | 28.8 | 35.6 |
| LightRAG (local) (GPT-4o-mini) | 28.0 | 30.0 | 20.6 | 26.2 | 34.2 | 30.8 | 24.0 | 29.7 | 18.6 | 19.5 | 9.4 | 15.8 | 32.0 | 27.6 | 16.0 | 25.2 |
| LightRAG (global) (GPT-4o-mini) | 27.6 | 27.3 | 18.2 | 24.4 | 36.0 | 24.7 | 16.7 | 26.1 | 18.4 | 16.2 | 6.4 | 13.7 | 30.4 | 23.1 | 13.6 | 22.4 |
| KG²RAG (LLaMA3.1-8B-Instruct) | 21.4 | 25.4 | 18.0 | 21.6 | 29.4 | 19.3 | 11.4 | 20.0 | 13.8 | 18.7 | 10.2 | 14.2 | 9.6 | 13.3 | 8.0 | 10.3 |
| KG-Infused RAG (GPT-4o-mini) | **35.2** | **43.8** | **32.8** | **37.3** | **40.8** | **40.5** | **31.4** | **37.6** | **24.4** | **30.8** | **18.8** | **24.7** | 40.8 | **44.7** | **37.6** | **41.0** |

**Retriever and Generator.** We adopt Contriever-MS MARCO (Izacard et al., 2022) as the retriever for both entity and passage retrieval in all experiments, except that for certain KG-based baselines, we follow their official implementations. For generation, we employ a diverse set of language models spanning parameter scales, covering both open-source and proprietary settings. Specifically, we evaluate with the Qwen2.5 series (Yang et al., 2024) (3B, 7B, 14B), the LLaMA3.1 series (Grattafiori et al., 2024) (8B, 70B), and the GPT-4o-mini (Hurst et al., 2024).

### 4.4 IMPLEMENTATION DETAILS.

**Knowledge Retrieval.** For all non-KG-based baselines and our method, the default number of retrieved passages $k_p$ is set to 6. In both Vanilla-QE and KG-Infused RAG, the original query $q$ and the expanded query $q'$ each retrieve $k_p/2 = 3$ passages from the corpus. For GraphRAG and LightRAG, we retain their default top-$k$ retrieval configurations specified in their original implementations. A comprehensive summary of the retrieval configurations for all methods is provided in Appendix D.2.

**KG-Infused RAG Implementation Details.** Retrieval is initialized by selecting the top $k_e = 3$ entities from the KG. The maximum number of activation rounds is set to six. To control expansion and mitigate noise introduced by overly large subgraphs, we apply two constraints: (1) `MAX_ENTITIES_PER_ROUND`: Limits the number of new entities $\{e'\}$ introduced in each activation round, preventing excessive expansion and potential noise. (2) `MAX_TRIPLES_PER_ENTITY`: Restricts the number of triples retrieved per entity, reducing noise from entities with disproportionately large numbers of triples (e.g., countries or regions).

Table 3: **Plug-in experimental results (%) of Self-RAG and DeepNote.** KG-Infused RAG as a plug-in module is implemented using the DPO-trained LLaMA3.1-8B.

| Method | HotpotQA | | | | 2WikiMQA | | | | MuSiQue | | | | Bamboogle | | | |
|---|---|---|---|---|---|---|---|---|---|---|---|---|---|---|---|---|
| | Acc | F1 | EM | Avg | Acc | F1 | EM | Avg | Acc | F1 | EM | Avg | Acc | F1 | EM | Avg |
| Self-RAG (selfrag-llama2-7b) | 26.2 | **33.7** | **24.0** | **28.0** | 23.0 | 27.3 | 22.0 | 24.1 | 4.4 | 10.4 | 3.8 | 6.2 | 8.8 | 17.3 | 8.0 | 11.4 |
| + KG-Infused RAG (w/ 2-Round) | 26.6 | 32.1 | 22.8 | 27.2 | **35.2** | **39.6** | **30.0** | **34.9** | **9.4** | **15.8** | **8.8** | **11.3** | **27.2** | **32.8** | **24.8** | **28.3** |
| DeepNote (LLaMA3.1-8B-Instruct) | 41.6 | **49.3** | **37.8** | 42.9 | 38.6 | 37.6 | 28.8 | 35.0 | 10.6 | 16.9 | 7.4 | 11.6 | **32.8** | **40.7** | 29.6 | **34.4** |
| + KG-Infused RAG (w/ 2-Round) | **43.4** | 49.0 | 37.6 | **43.3** | **46.2** | **47.5** | **37.0** | **43.6** | **12.8** | **19.0** | **9.4** | **13.7** | **32.8** | 39.8 | **30.4** | 34.3 |

## 5 RESULTS AND ANALYSIS

### 5.1 MAIN RESULTS

Table 1 presents the overall performance of our method across five datasets, while Table 2 presents a comparison against other KG-based methods.

**Results on QA Tasks.**   As shown in Table 1, our method consistently outperforms all standard baselines *across different model families and sizes* on both multi-hop QA and commonsense QA tasks. For example, with two rounds of activation, it achieves improvements of up to 18.7% across LLaMA3.1 models. These results demonstrate the effectiveness of activation-guided KG retrieval in supporting multi-step reasoning. By contrast, Vanilla QE shows little benefit, underscoring the importance of structured knowledge for complex tasks. Overall, the results demonstrate that KG-based query expansion and knowledge augmentation are essential for enhancing both multi-hop and commonsense QA, enabling models to capture richer context and reason more reliably.

**Effect of DPO Training.**   Although the DPO training data comes from 2WikiMQA, we observe gains in both in-domain and out-of-domain settings. For example, LLaMA3.1-8B-Instruct achieves a 9.4% improvement on 2WikiMQA, while also showing smaller gains on other datasets. These results suggest that even limited preference tuning can substantially enhance the model's ability to integrate KG facts, thereby improving the overall performance of our framework.

**Comparison with KG-Based Methods.**   We further compare with representative KG-based RAG methods that construct ad-hoc KGs via LLMs. In line with prior work, we adopt a smaller corpus for evaluation, as constructing KGs from large-scale corpora incurs prohibitive costs and is practically infeasible. Even under this restricted setting, KG construction remains inefficient (see Appendix H.2 for statistics). In contrast, our method leverages pre-existing KGs, thereby eliminating construction overhead. Beyond reducing preparation cost, it also delivers faster inference—9.39× compared to GraphRAG (global) and 1.55× compared to GraphRAG (local)—while achieving superior accuracy across multiple datasets. Taken together, these benefits yield a favorable efficiency–performance trade-off, making our approach practical for large-corpus QA scenarios.

### 5.2 PLUG-IN RESULTS

As shown in Table 3, integrating KG-Infused RAG into Self-RAG and DeepNote equips these methods with the ability to retrieve and reason over external KG facts, thereby strengthening their multi-hop reasoning. On 2WikiMQA, for example, Self-RAG combined with our method improves the Avg score by 10.8%, while DeepNote achieves an 8.6% gain. These results demonstrate the flexibility and generalizability of our activation-based KG retrieval module, which effectively enhances non-KG-based RAG pipelines. Additional experiments with Self-Ask are reported in Appendix H.1.

### 5.3 ABLATION STUDY

We evaluate the effectiveness of KG-guided spreading activation by ablating its two downstream modules, KG-Based QE and KG-Aug Gen. Both depend on the constructed subgraph summary $\mathcal{S}_{\mathcal{G},q}^{act}$.

As shown in Table 4, KG-Infused RAG outperforms the variant without KG-Based QE in most cases, indicating that spreading activation enhances retrieval by injecting structured knowledge into the query, thereby improving answer generation. Likewise, removing KG-Aug Gen leads to notable

Table 4: **Ablation study results (%).** Results are based on the DPO-trained LLaMA3.1-8B-Instruct.

| Method | HotpotQA | | | | 2WikiMQA | | | | MuSiQue | | | | Bamboogle | | | |
|---|---|---|---|---|---|---|---|---|---|---|---|---|---|---|---|---|
| | Acc | F1 | EM | Avg | Acc | F1 | EM | Avg | Acc | F1 | EM | Avg | Acc | F1 | EM | Avg |
| KG-Infused RAG (w/2-Round) | **36.6** | **43.8** | **31.0** | **37.1** | **45.0** | **46.8** | **37.0** | **42.9** | **12.2** | **18.5** | **8.8** | **13.2** | 31.2 | 33.6 | 24.8 | 29.9 |
| w/o KG-Based QE | 36.0 | 41.9 | 29.8 | 35.9 | 42.6 | 45.8 | 35.4 | 41.3 | 10.8 | 16.2 | 7.0 | 11.3 | **32.8** | **38.5** | **29.6** | **33.6** |
| w/o KG-Aug Gen | 36.2 | 42.9 | **31.0** | 36.7 | 34.6 | 36.9 | 28.8 | 33.4 | 9.6 | 15.8 | 6.2 | 10.5 | 25.6 | 30.2 | 22.4 | 26.1 |
| w/o Both | 34.6 | 41.8 | 30.6 | 35.7 | 31.6 | 34.2 | 27.4 | 31.1 | 6.4 | 12.6 | 4.0 | 7.7 | 21.6 | 28.8 | 20.0 | 23.5 |

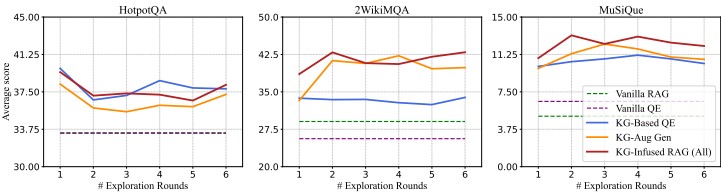

Figure 2: **Impact of the number of activation rounds on KG-Infused RAG.** Vanilla RAG and Vanilla QE use LLaMA3.1-8B-Instruct, while others use the DPO-trained version.

performance drops, suggesting that structured knowledge from spreading activation supplements retrieved passages and improves factual grounding. The improvements, however, are not uniform and may vary with the baseline retrieval performance and the alignment between the KG and the dataset.

## 5.4 ANALYSIS

**Evaluation of KG-Based QE: From a Retrieval Perspective.** We evaluate the effectiveness of KG-Based QE by comparing Vanilla RAG with and without KG-based activation. For Vanilla RAG, the top-6 passages are retrieved using only the raw query $q$. For Vanilla RAG + KG-Based QE, three passages are retrieved with $q$ and three with the expanded query $q'$ enriched with KG facts.

Table 5: **Retrieval performance** (%) on four datasets.

| Method | Hotpot | 2Wiki | MuSiQue | Bambo |
|---|---|---|---|---|
| | R@6 | R@6 | R@6 | R@6 |
| Vanilla RAG | 41.0 | 34.4 | 13.4 | 20.8 |
| + KG-Based QE (1-Round) | 45.0 | 40.0 | 19.4 | **24.8** |
| + KG-Based QE (2-Round) | **47.0** | **41.0** | **21.8** | **24.8** |

Table 5 shows that KG-Based QE significantly improves retrieval performance, with gains ranging from 4.0% to 8.4% across four datasets. These results confirm that activation-guided query expansion enables more relevant passage retrieval, laying a stronger foundation for downstream reasoning.

**Impact of activation rounds.** We examine the effect of activation depth, allowing up to six rounds, where KG-Based QE and KG-Aug Gen denote adding each component individually on top of Vanilla RAG. As shown in Figure 2, 2WikiMQA and MuSiQue achieve clear gains in the second activation round, though performance fluctuates thereafter, while HotpotQA peaks at the first round and then slightly declines. These results suggest that additional activation rounds do not guarantee improvements; limiting to one or two rounds is often both efficient and effective. Moreover, we observe that KG-Aug Gen contributes more than KG-Based QE on 2WikiMQA and MuSiQue, whereas on HotpotQA, KG-Based QE alone performs better. A possible explanation is that the effectiveness depends on the alignment between the dataset and the KG, as well as the LLM's ability to exploit structured knowledge.

## 6 CONCLUSION

In this work, we address limitations of existing RAG methods, which either rely solely on text corpora and overlook structural knowledge, or construct ad-hoc KGs with high cost and low reliability. To overcome these issues, we propose **KG-Infused RAG**, a framework that leverages pre-existing large-scale KGs and applies spreading activation across retrieval and generation. Experiments show that KG-Infused RAG achieves superior efficiency and performance compared with both vanilla RAG and KG-based methods, and it can also serve as a plug-in to strengthen corpus-based RAG methods such as Self-RAG. These results underscore both the effectiveness and flexibility of our method.

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

## A    STATEMENT ON LLM USAGE

In accordance with the ICLR 2026 policy on the use of large language models (LLMs), we disclose that LLMs (GPT) were employed solely for grammar correction and polishing of writing.

## B    ILLUSTRATIONS OF RETRIEVAL AND ACTIVATION PROCESSES

### B.1    COMPARISON OF RETRIEVAL PROCESSES

Figure 3 illustrates the differences in retrieval strategies between our KG-Infused RAG framework and existing RAG methods.

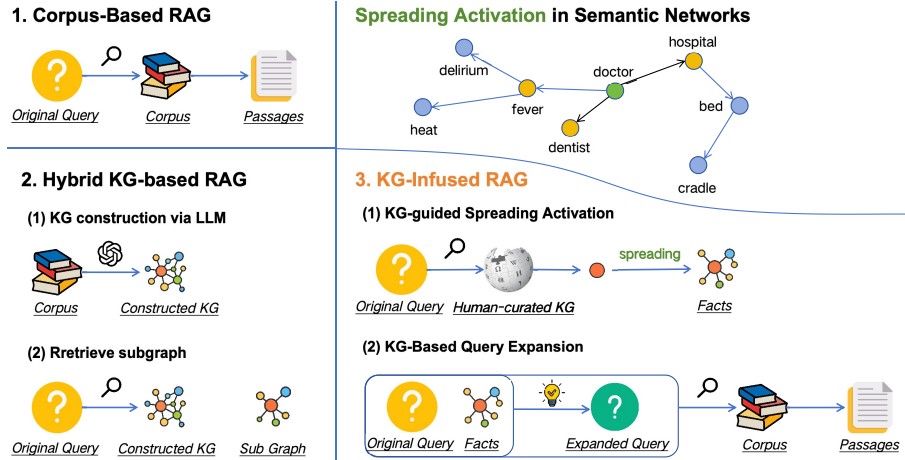

Figure 3: **Comparison of retrieval processes**: KG-Infused RAG vs. Existing RAG Methods.

### B.2    A CASE OF THE ACCUMULATED SUBGRAPH FROM SPREADING ACTIVATION

Figure 4 illustrates a case of the accumulated subgraph $\mathcal{G}_q^{act}$ in Wikidata5M-KG.

## C    KNOWLEDGE GRAPH CONSTRUCTION AND PREPROCESSING

### C.1    KG CONSTRUCTION: DATA PREPARATION

We prepare our KG $\mathcal{G}$ by extracting and preprocessing triples from Wikidata5M (Wang et al., 2021), a large-scale knowledge graph dataset derived from Wikidata and Wikipedia. Wikidata5M spans multiple domains, making it suitable for open-domain QA tasks. The KG used in KG-Infused RAG is defined as:

$$\mathcal{G} = \{\langle e, r, d\rangle \mid e \in \mathcal{E}, r \in \mathcal{R}, d \in \mathcal{D}_e\} \tag{12}$$

where $\mathcal{E}$, $\mathcal{R}$, and $\mathcal{D}_e$ denote the set of entities, relations, and entity descriptions, respectively. We preprocess Wikidata5M to meet our requirements for the KG, resulting in Wikidata5M-KG with approximately 21 million triples.

### C.2    PREPROCESSING DETAILS

To ensure the effectiveness of the KG-Guided Spreading Activation, we enforce two constraints on the preprocessed $\mathcal{G}$:

- **Entity Description Completeness.** Each entity $e$ must have a textual description $d$, which can be vectorized for similarity computation with the query.
- **Entity Triples Completeness.** Each entity $e$ must appear as the head entity in at least one triple to enable spreading activation starting from it.

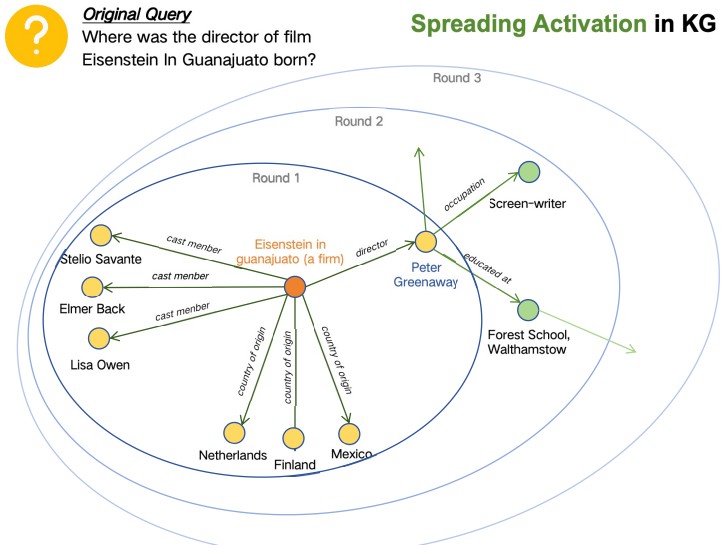

Figure 4: **An example of the accumulated subgraph $\mathcal{G}_q^{act}$ from spreading activation in Wikidata5M-KG.** This case, taken from the 2WikiMQA dataset, illustrates the resulting subgraph $\mathcal{G}_q^{act}$ constructed through KG-guided spreading activation. Due to space limitations, we only show part of the accumulated subgraph, and some activated entities are omitted from the figure.

We filter Wikidata5M by removing 5.65% of entities that lack descriptions or associated triples, resulting in a refined KG, denoted as Wikidata5M-KG. In contrast to constructing task-specific KGs with LLMs, using Wikidata5M-KG offers richer information, as it involves more entities and relationships, while also being more efficient, as it eliminates the need for LLM-based generation.

## C.3   STATISTICS OF KG

The statistics of Wikidata5M-KG are shown in Table 6. During preprocessing, some entities and relations were removed for failing to meet two constraints: **Entity Description Completeness** and **Entity Triples** Completeness. Specifically, 5.65% of entities and 2.17% of relations in the original Wikidata5M were discarded. As a result of these removals, only 1.72% of the triples were lost, indicating that the overall triple loss from the original dataset is minimal.

Table 6: **Statistics of Wikidata5M dataset and Wikidata5M-KG.**

| Dataset/KG | # Entity | # Relation | # Triple |
|---|---|---|---|
| Wikidata5M | 4944931 | 828 | 21354359 |
| Wikidata5M-KG | 4665331 | 810 | 20987217 |

## D   EXPERIMENTAL SETUP DETAILS

### D.1   DATASETS & EVALUATION

We summarize all datasets and the evaluation details in Table 7.

### D.2   MODEL BACKBONE & RETRIEVAL SETTING

Table 8 summarizes the model backbone and the retrieval setting used throughout our experiments. For LightRAG, top-$k$ denotes the number of tokens in the original chunks. For GraphRAG (local), the default configuration retrieves 10 entities and 10 relationships, while GraphRAG (global) adopts a

Table 7: **Summary of the properties of datasets used in experiments.**

| Property | HotpotQA | 2WikiMQA | MuSiQue | Bamboogle | StrategyQA |
|---|---|---|---|---|---|
| # Samples | 500 | 500 | 500 | 125 | 500 |
| Eval Metric | Acc,F1,EM | Acc,F1,EM | Acc,F1,EM | Acc,F1,EM | Acc |

map-reduce strategy over all relevant communities without a fixed top-$k$ limit. For the other methods, top-$k$ refers to the number of retrieved passages or chunks.

Table 8: **Overview of methods, backbones, and Top-k values.**

| Method | Model Backbone | Top-$k$ |
|---|---|---|
| Vanilla RAG | Qwen2.5 / LLaMA3.1 / GPT-4o series | 6 |
| Vanilla QE | Qwen2.5 / LLaMA3.1 / GPT-4o series | 6 |
| Self-Ask | Qwen2.5 | 6 |
| Self-RAG | Qwen2.5 / LLaMA3.1 | 6 |
| DeepNote | Qwen2.5 / LLaMA3.1 | 6 |
| LightRAG (local) | GPT-4o-mini | 60 |
| LightRAG (global) | GPT-4o-mini | 60 |
| GraphRAG (local) | GPT-4o-mini | 20 |
| GraphRAG (global) | GPT-4o-mini | / |
| KG$^2$RAG | LLaMA3.1 | 6 |
| KG-Infused RAG | Qwen2.5 / LLaMA3.1 / GPT-4o series | 6 |

## E PROMPT DETAILS

### E.1 PROMPTS FOR INFERENCE

Prompts enclosed in a black frame represent the base prompts used across different baselines or methods in the experiments. Prompts enclosed in a cyan frame indicate prompts specifically designed for or used within KG-Infused RAG.

Table 9: **Prompt of the answer generation stage (without retrieval).**

---

**Prompt of the Answer Generation Stage (without retrieval)**

**Instructions**
Only give me the answer and do not output any other words.
- - - - - - - - - - - - - - - - - - - - - - - - - - - - - - - - - - - - - - - - - - - - - - - - - - - -
Question: {question}
Answer:

---

Table 10: **Prompt of the answer generation stage (with retrieval).**

---

**Prompt of the Answer Generation Stage (with retrieval)**

**Instructions**
Answer the question based on the given passages. Only give me the answer and do not output any other words.
- - - - - - - - - - - - - - - - - - - - - - - - - - - - - - - - - - - - - - - - - - - - - - - - - - - -
Passages:
{passages}

Question: {question}
Answer:

---

Table 11: **Prompt of the vanilla query expansion.**

**Prompt of the Vanilla Query Expansion**

**Instructions**
Generate a new short query that is distinct from but closely related to the original question. This new query should aim to retrieve additional passages that fill in gaps or provide complementary knowledge necessary to thoroughly address the original question. Ensure the new query is relevant, precise, and broadens the scope of information tied to the original question. Only give me the new short query and do not output any other words.
- - - - - - - - - - - - - - - - - - - - - - - - - - - - - - - - - - - - - - - - - - - - - - - - - - -
Original Question:
{question}

New Query:

Table 12: **Prompt of the passage note construction stage.**

**Prompt of the Passage Note Construction Stage**

**Instructions**
Based on the provided document content, write a note. The note should integrate all relevant information from the original text that can help answer the specified question and form a coherent paragraph. Please ensure that the note includes all original text information useful for answering the question.
- - - - - - - - - - - - - - - - - - - - - - - - - - - - - - - - - - - - - - - - - - - - - - - - - - -
Question to be answered:
{question}

Document content:
{passages}

Note:

Table 13: **Prompt of the triples selection stage.**

**Prompt of the Triples Selection Stage**

**Instructions**
Given a question and a set of retrieved entity triples, select only the triples that are relevant to the question.

Information:
1. Each triple is in the form of <subject, predicate, object>.
2. The objects in the selected triples will be further explored in the next steps to gather additional relevant triples information.

Rules:
1. Only select triples from the retrieved set. Do not generate new triples.
2. A triple is relevant if it contains information about entities or relationships that are important for answering the question, either directly or indirectly.
    – For example, if the question asks about a specific person, include triples about that person's name, occupation, relationships, etc.
    – If the question asks about an event or entity, include related background information that can help answer the question.
3. Output triples exactly as they appear in angle brackets (<...>).
- - - - - - - - - - - - - - - - - - - - - - - - - - - - - - - - - - - - - - - - - - - - - - - - - - -
Question:
{question}

Retrieved Entity Triples:
{triples}

Selected Triples:

Table 14: **Prompt of the triples update stage.**

---

**Prompt of the Triples Update Stage**

**Instructions**
Given a question, a set of previously selected entity triples that are relevant to the question, and a new set of retrieved entity triples, select only the triples from the new set of retrieved entity triples that expand or enhance the information provided by the previously selected triples to help address the question.

Information:
1. Each triple is in the form of <subject, predicate, object>.
2. The objects in the selected triples will be further explored in the next steps to gather additional relevant triples information.

Rules:
1. Only select triples from the new set of retrieved entity triples. Do not include duplicates of the previously selected triples or generate new triples.
2. A triple is considered relevant if it:
   – Provides new information that complements or builds upon the entities, relationships, or concepts in the previously selected triples, and
   – Helps to better address or provide context for answering the question.
3. Do not include triples that are unrelated to the question or do not expand on the previously selected triples.
4. Output triples exactly as they appear in angle brackets (<...>).
------------------------------------------------------------------
Question:
{question}

Previously Selected Triples:
{previous_selected_triples}

New Retrieved Entity Triples:
{new_retrieved_triples}

Selected Triples:

---

Table 15: **Prompt of the triples (subgraph) summary stage.**

---

**Prompt of the Triples (Subgraph) Summary Stage**

**Instructions**
Given a question and a set of retrieved entity triples, write a summary that captures the key information from the triples. If the triples do not provide enough information to directly answer the question, still summarize the information provided in the triples, even if it does not directly relate to the question. Focus on presenting all available details, regardless of their direct relevance to the query, in a concise and informative way.
------------------------------------------------------------------
Question:
{question}

Selected Triples:
{selected_triples}

Summary:

---

Table 16: **Prompt of the KG-based query expansion stage.**

| Prompt of the KG-Based Query Expansion Stage |
| --- |
| **Instructions** 
 Generate a new short query that is distinct from but closely related to the original question. This new query should leverage both the original question and the provided paragraph to retrieve additional passages that fill in gaps or provide complementary knowledge necessary to thoroughly address the original question. Ensure the new query is relevant, precise, and broadens the scope of information tied to the original question. Only give me the new short query and do not output any other words. 
 ------------------------------------------------------------- 
 Original Question: 
 {question} 

 Related Paragraph: 
 {triples_summary} 

 New Query: |

Table 17: **Prompt of the KG-guided knowledge augmentation stage.**

| Prompt of the KG-Guided Knowledge Augmentation Stage |
| --- |
| **Instructions** 
 You are an expert in text enhancement and fact integration. Given a question, a retrieved passage, and relevant factual information, your task is to improve the passage by seamlessly incorporating useful details from the factual information. Ensure that the enhanced passage remains coherent, well-structured, and directly relevant to answering the question. Preserve the original meaning while making the passage more informative. Avoid introducing unrelated content. 
 ------------------------------------------------------------- 
 Question: 
 {question} 

 Retrieved Passage: 
 {passage} 

 Relevant Factual Information: 
 {triples_summary} 

 Enhanced passage: |

## E.2 PROMPTS FOR DPO DATA COLLECTION

Table 18: **Prompt of the KG-guided knowledge augmentation stage for DPO.**

---

**Prompt of the KG-Guided Knowledge Augmentation Stage for DPO**

**Instructions**
Task: You will receive a list of enhanced passage outputs generated based on a given question, a retrieved passage, and relevant factual information (triples summary).
Your task is to evaluate and compare the outputs to identify the best and worst ones.

Rules:
1. Focus only on the final enhanced passage. Ignore any prefatory comments, explanations, or formatting differences that do not affect content.
2. The quality of an enhanced passage is determined by:
    – Integration: How well the factual information has been integrated into the passage.
    – Coherence: The passage should be logically structured, readable, and maintain a natural flow.
    – Relevance: The enhanced passage should directly support answering the question.
    – Accuracy: Factual information should be incorporated correctly without hallucination or distortion.
    – Preservation: The original passage's meaning should be preserved and enhanced, not changed incorrectly.
3. If two outputs have substantially the same informational content (even if wording differs slightly), they are considered of equal quality.
4. If all outputs are of similar quality, or if no significant difference can be determined, use the same _id for both best and worst.
-------------------------------------------------------------------
Input:
Question:
{question}

Retrieved Passage:
{passage}

Relevant Factual Information:
{facts}

Enhanced Passage Outputs:
{output}

Output format:

Output the result as a JSON object:
json {{"best_id": <_id of the highest-quality output>, "worst_id": <_id of the lowest-quality output>}}

Important:
Do not include any explanations, just the JSON output.

## E.3 PROMPTS OF SELF-ASK

Table 19: **Prompt of Self-Ask.**

---

**Prompt of Self-Ask**

**Instructions**
Instruction: Answer through sequential questioning. Follow these rules:
1. Generate ONLY ONE new follow-up question per step
2. Each follow-up MUST use information from previous answers
3. NEVER repeat any form of follow-up question
4. When sufficient data is collected, give the final answer.

Format Template:
Follow up: [New specific question based on last answer]
Intermediate answer: [Concise fact from response] ... (repeat until conclusion)
So the final answer is: [Answer of the Original Question, only give me the answer and do not output any other words.]
- - - - - - - - - - - - - - - - - - - - - - - - - - - - - - - - - - - - - - - - - - - - - - - - -
**Few-shot demonstrations**
Question: Who lived longer, Muhammad Ali or Alan Turing?
Are follow up questions needed here: Yes.
Follow up: How old was Muhammad Ali when he died?
Intermediate answer: Muhammad Ali was 74 years old when he died.
Follow up: How old was Alan Turing when he died?
Intermediate answer: Alan Turing was 41 years old when he died.
So the final answer is: Muhammad Ali

Question: When was the founder of craigslist born?
Are follow up questions needed here: Yes.
Follow up: Who was the founder of craigslist?
Intermediate answer: Craigslist was founded by Craig Newmark.
Follow up: When was Craig Newmark born?
Intermediate answer: Craig Newmark was born on December 6, 1952.
So the final answer is: December 6, 1952

Question: Are both the directors of Jaws and Casino Royale from the same country?
Are follow up questions needed here: Yes.
Follow up: Who is the director of Jaws?
Intermediate answer: The director of Jaws is Steven Spielberg.
Follow up: Where is Steven Spielberg from?
Intermediate answer: The United States.
Follow up: Who is the director of Casino Royale?
Intermediate answer: The director of Casino Royale is Martin Campbell.
Follow up: Where is Martin Campbell from?
Intermediate answer: New Zealand.
So the final answer is: No

Question: Who was the maternal grandfather of George Washington?
Are follow up questions needed here: Yes.
Follow up: Who was the mother of George Washington? Intermediate answer: The mother of George Washington was Mary Ball Washington.
Follow up: Who was the father of Mary Ball Washington?
Intermediate answer: The father of Mary Ball Washington was Joseph Ball.
So the final answer is: Joseph Ball
- - - - - - - - - - - - - - - - - - - - - - - - - - - - - - - - - - - - - - - - - - - - - - - - -
Question: {question}

---

Table 20: **Statistics of DPO training data.** $D_{\text{KA}}^1$ and $D_{\text{KA}}^2$ denote the sampled examples from the first and second rounds of knowledge augmentation, respectively. $D_{\text{KA}} = D_{\text{KA}}^1 \cup D_{\text{KA}}^2$.

|  | $D_{\text{KA}}^1$ | $D_{\text{KA}}^2$ | $D_{\text{KA}}$ |
|---|---|---|---|
| # Sample for LLaMA3.1-8B | 1449 | 1460 | 2909 |
| # Sample for Qwen2.5-7B | 1202 | 1259 | 2461 |

## F  DPO TRAINING DETAILS

**DPO Data Construction.** We use an advanced LLM, GPT-4o-mini, to construct the training dataset $D_{\text{KA}}$ for DPO training. Specifically, we sample examples from the 2WikiMultiHopQA (Ho et al., 2020) training set and collect intermediate outputs from the KG-Guided KA stage. For each input $x$, we generate six candidate outputs using diverse decoding strategies, varying both the temperature and top-$p$ parameters. The sampling configurations used are:

$(\texttt{temperature, top-}p) = \{(0.0, 1.0), (0.3, 0.95), (0.5, 0.9), (0.7, 0.9), (0.9, 0.8), (1.0, 0.7)\}.$

We prompt GPT-4o-mini to identify the best and worst outputs among the candidates for each input. After filtering out low-confidence or ambiguous judgments, we construct preference-labeled triples $(x, y^-, y^+)$ to form the dataset for DPO:

$$D_{\text{DPO}} = \left\{ (x, y^-, y^+) \mid (x, y^-, y^+) \in D_{\text{KA}}, y^+ \succ y^- \right\} \tag{13}$$

The DPO training set is constructed from the first and second rounds of spreading activation, sampling 1500 examples from each round, resulting in over 3000 input instances. The resulting dataset is used to train the DPO models for LLaMA3.1-8B (2909 examples) and Qwen2.5-7B (2461 examples). The statistics of the training data are shown in Table 20.

**Hyper-parameters for Training.** During DPO training, we perform full parameter fine-tuning on 4×A800 GPUs, training the model for one epoch. The detailed hyper-parameters are shown in Table 21.

Table 21: **DPO training hyper-parameters** for LLaMA3.1-8B and Qwen2.5-7B.

| Parameter | LLaMA/Qwen |
|---|---|
| *General* | |
| Max sequence length | 8000 |
| Batch size | 4 |
| Learning rate | 5e-7 |
| Training epochs | 1 |
| Optimizer | AdamW |
| AdamW $\beta_1$ | 0.9 |
| AdamW $\beta_2$ | 0.999 |
| AdamW $\epsilon$ | 1e-8 |
| Weight decay | 0.0 |
| Warmup ratio | 0.0 |
| *DPO-Specific* | |
| DPO $\beta$ | 0.1 |
| Ref model mixup $\alpha$ | 0.6 |
| Ref model sync steps | 512 |

## G  DETAILS OF PLUG-IN BASELINE INTEGRATION

**Self-RAG** For Self-RAG, we follow the official setting, in which multiple passages are retrieved once and subsequently used for short-form QA tasks. We replace the original retrieval results with

Table 22: **Cost of KG construction and inference** (Total tokens = Input + Output).

| KG Construction | | | | |
|---|---|---|---|---|
| Method | Input tokens | Output tokens | Total tokens | Time |
| GraphRAG | – | – | – | 11,853s |
| LightRAG | 141,900,133 | 26,337,116 | 168,237,249 | 31,905s |
| KG$^2$RAG | 4,029,418 | 985,660 | 5,015,078 | 6,105s |
| KG-Infused RAG | / | / | / | / |
| Inference | | | | |
| Method | Input tokens | Output tokens | Total tokens | Time |
| GraphRAG (local) | 4,469.97 | 9.56 | 4,479.53 | 5.38s |
| GraphRAG (global) | 282,222.93 | 2,449.92 | 284,672.85 | 32.69s |
| LightRAG (local) | 17,397.29 | 71.28 | 17,468.57 | 4.41s |
| LightRAG (global) | 12,281.29 | 80.09 | 12,361.38 | 3.49s |
| KG$^2$RAG | – | – | – | 4.31s |
| KG-Infused RAG (GPT-4o-mini) | 3,368.52 | 558.28 | 3,926.80 | 3.48s |

those obtained via KG-Based QE after KG-Guided Spreading Activation, and then apply KG-Aug Gen to each passage individually. For generation, we employ `selfrag-llama2-7b`[2] as the generator, which is additionally fine-tuned following the original Self-RAG paper.

**DeepNote**   For DeepNote, since both its query refinement and our KG-Based QE serve the purpose of generating new queries, we preserve its original refinement step to maintain methodological integrity. Instead, we perform KG-Guided Spreading Activation to obtain the KG subgraph summary $\mathcal{S}_{\mathcal{G},q}^{act}$, followed by KG-Guided KA to construct a fact-enhanced note $\mathcal{S}_q^{final}$, which is subsequently used for answer generation.

**Self-Ask.**   For Self-Ask, to ensure consistent settings and reduce API costs, we replace the original web-based search engine with our local corpus retriever and substitute the GPT-3 generator with the Qwen2.5-7B. The Self-Ask baseline generates one sub-question ("follow up" question) iteratively based on few-shot demonstrations and the previous reasoning process, then retrieves passages for the sub-question and generates a corresponding answer. This continues until the model determines it can produce the final answer. **To incorporate KG-Infused RAG as a plug-in module**, we enhance the sub-question answering stage. Specifically, we first perform KG-Based QE on each sub-question to retrieve new passages, followed by KG-Aug Gen to enhance the retrieved passages with relevant KG triples. Answers are then generated based on the fact-enhanced notes.

# H   ADDITIONAL EXPERIMENTAL RESULTS

## H.1   EXPERIMENTAL RESULT OF SELF-ASK

KG-Infused RAG is compatible with various corpus-based RAG frameworks. Here, we integrate it into Self-Ask and evaluate its effectiveness.

As shown in Table 23, combining KG-Infused RAG with Self-Ask yields moderate gains on four datasets, while a slight performance drop is observed on HotpotQA. Compared to Self-RAG, the overall improvements are smaller and less consistent. This difference is likely due to Self-Ask's step-by-step decomposition of complex questions into simpler single-hop sub-questions (see prompt in Table 19 and examples in Table 26). This decomposition enables effective retrieval using only the corpus, thereby reducing reliance on external structured knowledge. In such cases, the marginal benefit of KG-based augmentation diminishes, and injecting external facts may introduce irrelevant information that interferes with reasoning.

## H.2   COMPUTATION COST ANALYSIS

Table 22 reports the token usage and time cost of different KG-based methods during both KG construction and inference. For GraphRAG and LightRAG, we adopt the default gpt-4o-mini model

---

[2]https://huggingface.co/selfrag/selfrag_llama2_7b

Table 23: **Plug-in experimental results (%) of Self-Ask.** KG-Infused RAG as a plug-in module is implemented using the DPO-trained LLaMA3.1-8B.

| Method | HotpotQA | | | | 2WikiMQA | | | | MuSiQue | | | | Bamboogle | | | | StrategyQA |
|---|---|---|---|---|---|---|---|---|---|---|---|---|---|---|---|---|---|
| | Acc | F1 | EM | Avg | Acc | F1 | EM | Avg | Acc | F1 | EM | Avg | Acc | F1 | EM | Avg | Acc |
| Self-Ask | **36.6** | **42.6** | **35.4** | **38.2** | 39.2 | 42.4 | 33.6 | 38.4 | 11.8 | 16.7 | 7.8 | 12.1 | 32.8 | 40.1 | 29.6 | 34.2 | 65.6 |
| + KG-Infused RAG (w/ 1-Round) | 34.2 | 40.3 | 31.8 | 35.4 | **44.8** | **45.9** | **35.2** | **42.0** | **13.2** | **18.9** | **10.8** | **14.3** | 32.0 | 38.5 | 28.0 | 32.8 | **69.6** |
| + KG-Infused RAG (w/ 2-Round) | 34.8 | 41.1 | 32.0 | 36.0 | 41.8 | 43.1 | 33.0 | 39.3 | 12.6 | 17.8 | 9.0 | 13.1 | 30.4 | 37.0 | 26.4 | 31.3 | 69.2 |
| + KG-Infused RAG (w/ 3-Round) | 35.6 | 41.6 | 33.0 | 36.7 | 40.6 | 41.3 | 30.8 | 37.6 | 12.4 | 18.0 | 9.2 | 13.2 | **33.6** | **40.5** | **30.4** | **34.8** | 68.2 |

for knowledge graph construction as well as inference. To measure the inference overhead, we present the average cost of each method on the MuSiQue dataset.

# I LIMITATION

**KG Data Modalities.**  This work focuses on activating structured knowledge in the form of triples from KGs, which we integrate into a corpus-based RAG pipeline. However, KGs often contain heterogeneous forms of information beyond triples, such as long-form text, images, and tables. Our current framework is designed specifically for triple-based activation and does not yet account for these heterogeneous modalities. Effectively leveraging such diverse KG content in the activation process is an important direction for future research, potentially enabling broader applicability and richer factual grounding.

**Triple Utilization.**  Our method leverages KG triples in two main ways: query expansion and knowledge augmentation of retrieved passages. While effective, we believe KG facts can be further exploited through other strategies to better align with corpus retrieval and enhance downstream reasoning. Investigating more flexible or task-specific integration mechanisms remains an open and valuable area of future work.

# J CASE STUDY

## J.1 CASE I OF MAIN RESULTS: DIRECT ANSWER RETRIEVAL VIA KG ACTIVATION

Table 24 presents a case study comparing the answer generation process of KG-Infused RAG and two baselines. This example illustrates how, through two rounds of KG-guided activation, KG-Infused RAG directly acquires answer-relevant triples from the KG, providing sufficient structured context for the subsequent stages.

KG-Infused RAG begins by retrieving key query-relevant triples from the KG during the KG-Guided Spreading Activation stage. These triples explicitly capture the relationship between the manufacturer of the AAM-N-4 Oriole and its successor company. Leveraging these structured facts, KG-Infused RAG performs KG-based query expansion, generating a targeted query that focuses on pivotal events and transitions in the history of Martin Marietta after it succeeded the Martin Company. In the subsequent KG-augmented generation stage, KG-Infused RAG integrates the retrieved facts into a fact-enhanced note, which clearly states the corporate succession and related developments, ultimately supporting an accurate and grounded final answer.

In contrast, the answer generation processes of the two baselines suffer from the lack of such structured knowledge input. Vanilla RAG mistakenly associates the missile "AAM-N-4 Oriole" with the unrelated "Oriole Records" label due to misleading surface-form overlaps in the corpus, leading to an erroneous inference that CBS followed the missile manufacturer. Meanwhile, Vanilla QE misinterprets the earlier development date of Ryan Aeronautical's missile project as indicating that Ryan "followed" the Glenn L. Martin Company, incorrectly equating temporal precedence with corporate succession.

In summary, KG-Infused RAG benefits significantly from its activation-based use of KG triples, which provide high-precision, structured context that guides the downstream stages toward factual correctness. Without access to such KG-derived evidence, the baselines rely solely on corpus-based retrieval, which often fails to retrieve passages from the corpus that are truly effective for answering the question, leading to factually incorrect answers.

Table 24: **An example from the MuSiQue dataset.** Blue text indicates segments that positively support answering the question, while red text marks segments that are either uninformative or provide incorrect information. The ✗ denotes a wrong answer, and ✓ represents a correct answer.

---

**Question:** Which company followed the company that made AAM-N-4 Oriole?

---

*Process of Vanilla RAG:*
**Passage note:** The company that followed the Glenn L. Martin Company, which developed the AAM-N-4 Oriole, is not directly related to the information provided in the document. ... However, if we look at the information provided in the document about the Oriole record label, we can see that it was acquired by CBS in 1964...

---

*Process of Vanilla QE:*
**New query:** What companies developed and produced the AAM-N-4 Oriole's successors or related air-to-air missiles in the United States?
**Passage note:** The company that followed the Glenn L. Martin Company, which developed the AAM-N-4 Oriole, was the Ryan Aeronautical Company. The Ryan Aeronautical Company developed the AAM-A-1 Firebird, an early American air-to-air missile, which was the first air-to-air missile program developed for the United States Air Force. The AAM-A-1 project began in 1946, a year before the development of the AAM-N-4 Oriole started in 1947. The Ryan Aeronautical Company's AAM-A-1 Firebird was extensively tested in the late 1940s but...

---

*Process of KG-Infused RAG:*
**Retrieved query-relevant triples (After 2-round KG-guided Spreading Activation):**
    `<rtv-n-16 oriole | manufacturer | martin company>`
    `<rtv-n-16 oriole | instance of | Air-to-air missiles>`
    `<martin company | followed by | Martin Marietta>`
**Triples summary:** ...Based on the provided information, it appears that Martin Marietta is the company that followed the company that made the AAM-N-4 Oriole, as Martin Marietta is the successor to the Martin Company.
**New query:** What were the key events and milestones in the history of Martin Marietta after it succeeded the Martin Company, particularly in relation to its involvement in the development and production of air-to-air missiles?
**Passages note (after query expansion):** ...The Glenn L. Martin Company developed the AAM-N-4 Oriole, an early American air-to-air missile, for the United States Navy. In 1961, the Martin Company merged with American-Marietta Corporation to form Martin Marietta Corporation, a leading company in chemicals, aerospace, and electronics. Martin Marietta continued to evolve and eventually merged with Lockheed Corporation in 1995 to form Lockheed Martin Corporation, becoming the world's largest defense contractor.
**Fact-enhanced note:** The company that followed the Glenn L. Martin Company, which developed the AAM-N-4 Oriole, an early American air-to-air missile for the United States Navy, is Martin Marietta Corporation. The Martin Company, a renowned manufacturer of the AAM-N-4 Oriole, an instance of Air-to-air missiles, merged with American-Marietta Corporation in 1961 to form Martin Marietta Corporation, a leading company in chemicals, aerospace, and electronics. Martin Marietta continued to evolve and eventually merged with Lockheed Corporation in 1995 to form Lockheed Martin Corporation, becoming the world's largest defense contractor.

---

**Vanilla RAG:** CBS ✗
**Vanilla QE:** Not mentioned ✗
**KG-Infused RAG:** Martin Marietta Corporation. ✓

---

**Golden Answer:** Martin Marietta

## J.2   CASE II OF MAIN RESULTS: KG-GUIDED QUERY SIMPLIFICATION VIA INTERMEDIATE NODES

Table 25: **Examples from the 2WikiMQA dataset.** We present the original multi-hop queries and the corresponding simplified queries generated through KG-based Query Expansion (2-round). Blue text indicates the simplified or key-focused portions in the expanded query compared to the raw query.

| |
|---|
| **Raw Query:** Where was the director of film Eisenstein In Guanajuato born? |
| **Expanded Query:** What is Peter Greenaway's birthplace? |
| **Raw Query:** What is the place of birth of Elijah Blue Allman's father? |
| **Expanded Query:** What is the place of birth of Gregg Allman? |
| **Raw Query:** What is the date of birth of George Frederick, Count Of Erbach-Breuberg's father? |
| **Expanded Query:** What are the dates of birth and death of George Albert I, Count of Erbach-Schonberg's siblings? |
| **Raw Query:** Which film whose director is younger, Many Tanks Mr. Atkins or Do Musafir? |
| **Expanded Query:** What are the ages of the directors of the 2004 film Musafir and the 2013 film Musafir, and how do they compare to the age of R. William Neill? |

Even when the answer node itself is not present in the KG—making it impossible to directly retrieve the answer via multi-round KG-guided activation—the retrieved multi-hop intermediate nodes can still play a crucial role. Specifically, KG-guided Spreading Activation can still identify intermediate entities and relations that form partial reasoning chains, providing useful cues for query reformulation and retrieval. In such cases, KG-Based QE transforms the original complex question into one or more simplified single-hop sub-questions that focus on intermediate concepts surfaced by the KG. These simplified queries improve corpus retrieval by aligning better with the surface forms and local context found in text passages. Table 25 presents several examples of simplified single-hop questions generated by KG-Based QE, demonstrating how intermediate KG nodes help bridge the gap between the original query and corpus evidence.

## J.3  CASE OF SELF-ASK

Table 26: **Examples from HotpotQA and 2WikiMQA.** Blue text indicates segments that positively support answering the question, while red text marks segments that are either uninformative or provide incorrect information. The ✗ denotes a wrong answer, and ✓ represents a correct answer.

---

**Question 1 (from HotpotQA):** Emily Beecham is best known for her role in a televsion series whose second season premiered on what date?

---

*Reasoning Process of Self-Ask:*
{Prompt of Self-Ask, see in Table 19}
**Question:** Emily Beecham is best known for her role in a televsion series whose second season premiered on what date?
Are follow up questions needed here: **Follow up:** What is the name of the television series in which Emily Beecham starred?
**Intermediate answer:** Into the Badlands.**Follow up:**When did the second season of Into the Badlands premiere?
**Intermediate answer:** March 19, 2017.
**So the final answer is:**March 19, 2017 ✓

---

*Reasoning Process of Self-Ask + KG-Infused RAG:*
{Prompt of Self-Ask, see in Table 19}
**Question:** Emily Beecham is best known for her role in a televsion series whose second season premiered on what date?
Are follow up questions needed here: **Follow up:** What is the name of the television series in which Emily Beecham starred?
**Intermediate answer:** Afterlife.**Follow up:** When did the second season of Afterlife premiere?
**Intermediate answer:** 16 September 2006.
**So the final answer is:**16 September 2006 ✗

---

**Golden Answer:** March 19, 2017

---

**Question 2 (from 2WikiMQA):** Who is the child of the director of film Mukhyamantri (1996 Film)?

---

*Reasoning Process of Self-Ask:*
{Prompt of Self-Ask, see in Table 19}
**Question:** Where was the place of death of the director of film Happy Go Lucky (1936 Film)?
Are follow up questions needed here: Yes.
**Follow up**: Who was the director of the film Happy Go Lucky (1936 Film)?
**Intermediate answer**: The passage does not provide information about a 1936 film titled "Happy Go Lucky." The director for the 1943 film is given as Curtis Bernhardt, but there is no information about a 1936 version. Therefore, the answer to the question as posed cannot be determined from the given information.
**So the final answer is:**Information not sufficient to determine the place of death of the director of a 1936 film titled "Happy Go Lucky."✗

---

*Reasoning Process of Self-Ask + KG-Infused RAG:*
{Prompt of Self-Ask, see in Table 19}
**Question:** Where was the place of death of the director of film Happy Go Lucky (1936 Film)?
Are follow up questions needed here: Yes.
**Follow up:** Who was the director of the film Happy Go Lucky (1936 Film)?
**Intermediate answer:** Aubrey Scotto.**Follow up:** Where did Aubrey Scotto die?
**Intermediate answer:** Los Angeles.
**So the final answer is:**Los Angeles.✓

---

**Golden Answer:** Los Angeles

