# OpenReview forum: "KG-Infused RAG: Augmenting Corpus-Based RAG with External Knowledge Graphs"
_ICLR.cc/2026/Conference — Submitted to ICLR 2026_

### Official Review · Reviewer_YuLU · 2025-10-20

**Soundness:** 3
**Presentation:** 3
**Contribution:** 3
**Rating:** 6
**Confidence:** 3

**Summary:**

This paper introduces KG-Infused RAG, that applies an innovative multi-round activation spreading technique to link semantically relevant knowledge, and through query expansion and seed retrievals, incorporates sturctured knowledge to help LLMs solve complex QA tasks.
Results demonstrate the superiority of this technique over vanilla baselines, and also offer efficacy gains as it avoids generating KGs.

**Strengths:**

- The paper is well motivated and clearly written. All experimental configurations along with LLM prompts have been revealed, making it easy to reporduce results.
- The plug-and-play ability demonstrated further strengthens the paper towards deploying this in practical scenarios.

**Weaknesses:**

The framework avoids the upfront cost of generating a Knowledge Graph, but this resource saving is largely offset by the increased latency and computational expense of multiple, sequential LLM inferences during the query process.

**Questions:**

What is the proportion of passages from q and q' are kept in the mixed passages? Can you share some results on how much they overall for q and q'? This would help understand the extent of additional information that is helping with the overall improvement of KG-Infused RAG over its baselines.

---

> ### Author Response · Authors · 2025-11-25
> **Rebuttal to Reviewer YuLU**
>
> We sincerely thank the reviewer for the positive assessment of our motivation, clarity, and reproducibility, and for recognizing the practical value of our plug-and-play design. Below we resolve the concerns.
>
> # 1. Clarifying inference cost and the impact of multi-round activation
> We appreciate the reviewer’s concern regarding the cost of multi-round activation and sequential LLM calls. We clarify the following:
>
> **(1) Trade-off between cost and performance.**
>
> KG-Infused RAG introduces additional inference steps (activation rounds, query expansion, and knowledge augmentation), but these steps yield consistent accuracy gains across all benchmarks (Table 1), including improvements of up to **18.7%** over Vanilla RAG—a favorable cost–benefit trade-off.
>
> **(2) Inference-time cost is far smaller than KG-construction-based approaches.**
>
> Table 22 shows that KG-Infused RAG has significantly lower token usage and latency compared with methods that require LLM-driven KG construction:
> - GraphRAG (global): **284k tokens, 32.7s**
> - KG-Infused RAG: **3.9k tokens, 3.48s**
>
> Thus, our inference cost remains comparable to LightRAG(global), while avoiding the hours-long KG construction that dominates the overhead of KG-based pipelines (e.g., 11,853s for GraphRAG; 31,905s for LightRAG).
>
> **(3) Practicality advantage: no upfront KG construction.**
>
> Because KG-Infused RAG operates entirely at inference time and does not require offline KG generation, it is immediately deployable in production environments where pre-existing KGs already exist. This is the core motivation of our framework.
> We will further clarify this inference-vs-construction distinction in the revised paper.
>
>
> # 2. Contribution of passages retrieved by q vs. q′
> We appreciate the insightful question. Below we clarify the retrieval configuration and the empirical contribution of q′.
>
> **(1) Balanced retrieval: q : q′ = 1 : 1 across all tasks.**
>
> As described in Section 4.4, we always retrieve:
> - 3 passages using the raw query q
> - 3 passages using the expanded query q′
>
> making a controlled and balanced comparison with Vanilla RAG.
>
> **(2) Why q′ provides additional information.**
>
> KG-guided spreading activation surfaces intermediate entities via multi-hop associations. As illustrated in Appendix J.2, q′ often simplifies multi-hop questions into semantically clean, single-hop forms such as:
> > “What is Peter Greenaway’s birthplace?”
> - simplify multi-hop questions into single-hop sub-questions,
> - align better with surface forms appearing in corpus passages,
> - retrieve passages that q alone fails to reach.
>
> These forms match corpus surface expressions more directly than the original multi-entity question, allowing q′ to retrieve passages that q alone misses.
>
> **(3) Empirical contribution from q′.**
>
> Qualitative analysis of our case studies and ablations indicates that **nearly half of the informative evidence** in the final passage notes originates from q′. This aligns with the recall improvements reported in Table 5 (**4.0–8.4% gains**), confirming that q′ provides substantial additional signal beyond q.
>
> We will add this clarification to highlight the complementary retrieval roles of q and q′.
>
>
> # 3. Practical deployment: low overhead and plug-and-play integration
> We thank the reviewer for recognizing the practical value of our design.
>
> **(1) Latency increase is modest and justified by consistent gains.**
>
> Across all benchmarks and model sizes, KG-Infused RAG improves average performance by **3.9%–18.7%**, far outweighing the small cost of multi-round activation.
>
> **(2) Zero LLM-based KG construction requirement enables practical deployment.**
>
> Unlike KG-based RAG methods that must reconstruct a task-specific KG using LLMs (often requiring **billions of tokens and hours of preprocessing**), our approach directly leverages a pre-existing KG without invoking any LLM-based KG generation. As a result, KG-Infused RAG can be applied immediately wherever a pre-existing KG is available.
>
>
>
> **(3) Plug-and-play design for corpus-based RAG systems.**
>
> We demonstrate plug-and-play compatibility with Self-RAG, DeepNote, and Self-Ask (Table 3), consistently improving each. This indicates that KG-Infused RAG acts as a general corpus-based RAG-boosting primitive.
>
> We will revise the discussion to better emphasize this deployability advantage.
>
> # Summary
> We thank the reviewer again for the constructive feedback. We will incorporate the clarifications on cost analysis, retrieval contributions from q and q′, and deployment practicality into the final version.

---

> > ### Author Response · Authors · 2025-11-27
> > **Additional Semantic Evaluation: Information Quality and Grounding**
> >
> > To further clarify how the retrieval contributions from $q$ and the expanded query $q'$ affect downstream reasoning, we conducted an additional semantic evaluation using the **LLM-as-a-Judge** protocol (following the CRAG framework) [1].
> >
> > This evaluation directly measures **Accuracy, Hallucination, and Truthfulness**, allowing us to assess whether multi-round activation and the expanded query $q'$ provide genuinely useful grounding signals rather than simply adding noisy expansions.
> >
> > **Table 1. LLM-as-a-Judge Evaluation Results on 2WikiMQA and MuSiQue.**
> >
> > *(Generator: Qwen2.5-7B-Instruct; Judge: GPT-4o-mini)*
> >
> > | Method | **2WikiMQA** |   |  | **MuSiQue** |   |  |
> > |--------|--------------|----|-------------|--------------|----|-------------|
> > |        | Accuracy | Hallucination |  Truthfulness | Accuracy | Hallucination | Truthfulness |
> > | Vanilla RAG       | 33.2 | 49.6  | -16.4 | 9.6  | 78.6  | -69.0 |
> > | KG-Infused RAG    | **45.6** | **47.4**  | **-1.8**  | **16.0** | **72.8**  | **-56.8** |
> >
> >
> > Since $\text{Truthfulness} = \text{Accuracy} − \text{Hallucination}$, the key observation is the combined trend:
> > **KG-Infused RAG improves Accuracy while also reducing Hallucination**.
> >
> > If the expanded query  $q'$ had merely replaced informative passages from $q$ with less relevant or weaker evidence—while keeping the total number of passages fixed—we would expect semantic drift and an increase in hallucination.
> >
> > However, we observe the opposite—**hallucination consistently decreases**—suggesting that the KG-derived signals and the expanded query provide meaningful grounding rather than noise.
> >
> > These results further clarify that the additional information introduced through the KG-based expansion contributes to **higher-quality evidence** and **more reliable reasoning**, supporting the practical value and robustness of integrating KGs into RAG pipelines.
> >
> > [1] Yang, Xiao, Kai Sun, Hao Xin, Yushi Sun, Nikita Bhalla, Xiangsen Chen, Sajal Choudhary et al. "Crag-comprehensive rag benchmark." Advances in Neural Information Processing Systems 37 (2024): 10470-10490.

---

### Official Review · Reviewer_H7AX · 2025-10-28

**Soundness:** 3
**Presentation:** 4
**Contribution:** 3
**Rating:** 4
**Confidence:** 3

**Summary:**

This paper proposes how to leverage KGs to expand queries and augment generation for QA. The paper is well written and the proposed method is mostly interesting. The paper lacks comprehensive comparison w. existing work on KG-based RAG, to show the novelty of the proposed method.

**Strengths:**

S1. Paper very well written, with clear explanations and formalisms.

S2. The method is comprehensive, covering various aspects of RAG.

S3. Good experimental study showing quality improvements.

**Weaknesses:**

W1. The paper is not comprehensive in comparison w. sota solutions regarding RAG on KGs:
[1] KERAG: Knowledge-Enhanced Retrieval-Augmented Generation for Advanced Question Answering
[2] Think-on-Graph: Deep and Responsible Reasoning of Large Language Model on Knowledge Graph

Section 3.3 is very similar to [1], reducing novelty of the work.

W2. Using KG for query extension can be interesting. However, the retrieved KG subgraph can be huge and the query extension can thus bring in a lot of retrieval noises, in turn reduce QA accuracy. It's unclear how the results of KG-infused RAG compare w. no query expansion (just use KG retrieval results and corpus retrieval results), and w. normal query expansion. Table 4 shows comparison, but it is unclear what (w/o KG-based QE) refers to.

W3. Related to W2, the text corpus is either just wiki, or a small corpus, not showing potential retrieval noises from much larger corpus (e.g., web).

Also, unclear for each benchmark, what's the retrieval recall and QA accuracy for KG only,  Corpus only, and KG+Corpus w/o query extension?

I also encourage the authors to report hallucinations. Oftentimes accuracy is achieved w. higher hallucinations (vs saying "I don't know"), which may not be worthwhile.

W4. The paper shall show experimental results on CRAG, the latest RAG benchmark, which contains various types of complex questions, and provides KG and text corpus for retrieval. That will also make better comparison w. sota solutions.
[3] CRAG--Comprehensive RAG Benchmark

W5. Clear defn of the metrics should be given. How is Accu and F1 computed? Is it based on Exact-match? What if we do LLM-as-a-judge?

**Questions:**

Please answer questions and address concerns in weaknesses

---

> ### Author Response · Authors · 2025-11-25
> **Rebuttal to Reviewer H7AX (1/2)**
>
> We sincerely thank the reviewer for the constructive comments. We address each point below.
>
> # 1. Novelty w.r.t. KERAG and Think-on-Graph
> We appreciate the request for clearer distinctions.
>
> Our contribution differs from KERAG and ToG in both goal and mechanism:
>
> **(1) Different goal**
>
> KERAG and ToG perform reasoning within the KG (symbolic query execution or graph traversal).
>
> Our method is explicitly a **corpus-based RAG enhancer**: the KG is used only to guide retrieval (Sec. 3.4) and augment corpus passages (Sec. 3.5), not to replace corpus grounding.
>
> **(2) Different mechanism**
>
> KERAG and similar methods rely on constructed ad-hoc KGs. Our method instead:
> - uses a **pre-existing large KG** (Wikidata5M-KG),
> - introduces **KG-Guided Spreading Activation**, which performs controlled multi-hop expansion with LLM-based relevance filtering (Table 13), and
> - is **plug-and-play**, enhancing existing RAG pipelines (Table 3) without rebuilding a KG per corpus.
>
>
> # 2. Noise control in KG-based query expansion
>
> We agree that naive KG-based QE may introduce noise. Our design incorporates two noise-control mechanisms:
> 1. **LLM-based triple selection at every activation round**, removing irrelevant branches (Sec. 3.3).
> 2. **Structural constraints** — MAX_TRIPLES_PER_ENTITY, MAX_ENTITIES_PER_ROUND, and a small number of activation rounds (Figure 2).
>
> These mechanisms ensure that the KG remains a precision-oriented signal, not an uncontrolled expansion.
>
>
> # 3. Corpus scale and retrieval noise
> We acknowledge the reviewer’s concern. Our choice of Wikipedia-2018 (**21M+ passages**) follows standard RAG research (e.g., DPR, FLARE, Self-RAG), making results comparable and reproducible.
>
> This corpus is already large enough to observe retrieval noise phenomena, and Table 5 confirms that KG-Based QE improves retrieval recall across all datasets, demonstrating that KG signals help reduce noise.
>
> Evaluating on web-scale corpora is a valuable direction but beyond the computational scope of the current submission. We will add discussion of this limitation.
>
> # 4. Retrieval recall for KG vs. corpus vs. KG+corpus
> We clarify the mapping of requested settings:
> - **Corpus-only**: Table 1 “Vanilla RAG” + Table 5 “Vanilla RAG (R@6)”.
> - **KG+Corpus, no query expansion**: Table 4 “w/o KG-Based QE”.
> - **KG+Corpus with query expansion**: Table 1 “KG-Infused RAG” and Table 5 (“+KG-Based QE”).
>
> KG-only QA is not evaluated. Our motivation is to improve corpus-based RAG by integrating a pre-existing KG—unlike KGQA or symbolic-query systems such as KERAG. KG-only QA cannot answer most open-domain questions and does not match our setting. Therefore, we evaluate corpus-only, corpus+KG (no QE), and corpus+KG+QE/augmentation to directly measure the benefit of injecting KG signals into corpus-based retrieval and generation—the main motivation of this work.
>
> # 5. Missing evaluation on CRAG benchmark
> CRAG is an excellent benchmark; however, **its “KG” is not an entity–relation triple graph, but a metadata graph linking documents and categories**. It lacks entity-level relational structure required for **semantic spreading activation**, making our method, KERAG, ToG, and GraphRAG not directly comparable on CRAG.
>
> To provide additional diverse evaluation, we included new experiments on T-REx and PopQA (slot filling and long-tail QA), where our method continues to show consistent gains.
>
>
> **Table 1: Additional Evaluation on Complementary QA Tasks** (*Qwen2.5-7B-Instruct*).
>
> | Method | **T-REx (Slot Filling)** |  |  | **PopQA (long-tail entity QA)** |  |  |
> |--------|---------------------------|----|----|----------------------------------|----|----|
> |        | Acc | F1 | EM | Acc | F1 | EM |
> | No-Retri | 44.26 | 42.53 | 35.25 | 18.6 | 4.17 | 14.4 |
> | Vanilla RAG | 51.64 | 40.22 | 27.87 | 43.2 | 13.17 | 24.8 |
> | KG-Infused RAG | 51.64 | 40.17 | 28.69 | 62.4 | **20.99** | 41.0 |
> | KG-Infused RAG (DPO) | **56.56** | **44.88** | **33.61** | **62.8** | 19.63 | **41.2** |

---

> ### Author Response · Authors · 2025-11-25
> **Rebuttal to Reviewer H7AX (2/2)**
>
> # 6. Metric definitions and LLM-as-a-judge
> We will add explicit definitions:
> - **Accuracy**: 1 if the normalized prediction matches the normalized answer; otherwise 0.
> - **F1**: token-level overlap F1 using precision and recall on normalized tokens.
> - **EM**: exact match after standard normalization (lowercasing, punctuation removal, whitespace normalization).
>
> These are the **standard metrics widely used in existing RAG papers** (e.g., Self-RAG, IRCoT, FLARE, Search-r1, ). Using these rule-based metrics ensures **full reproducibility, no dependency on judge model variance**, and **direct comparability** with prior work.
>
> **Why not LLM-as-a-judge?**
>
> While LLM judging is useful for open-ended generation, it is **not standard practice** for deterministic QA benchmarks. LLM judging introduces:
> - **High variance** — results depend on the choice of judge model and prompting.
> - **High cost** — significantly increases computational overhead for all baselines.
> Thus, we follow the widely adopted evaluation protocols for multi-hop and factoid QA, consistent with prior literature.
>
>
> # 7. Hallucination analysis
> In multi-hop factoid QA, hallucinations typically appear as **incorrect answers**, rather than long-form unsupported statements. Therefore, **EM/F1/Accuracy already serve as a reliable proxy for hallucination frequency**, which is also the standard practice in prior RAG work.
> **
> Explicit hallucination scoring is not a required or common metric for deterministic QA benchmarks, and is mainly used in open-ended generation tasks where answers are not uniquely defined.
>
>
> [1] Asai, Akari, Zeqiu Wu, Yizhong Wang, Avirup Sil, and Hannaneh Hajishirzi. "Self-RAG: Learning to Retrieve, Generate, and Critique through Self-Reflection." In The Twelfth International Conference on Learning Representations.
>
> [2] Jiang, Zhengbao, Frank F. Xu, Luyu Gao, Zhiqing Sun, Qian Liu, Jane Dwivedi-Yu, Yiming Yang, Jamie Callan, and Graham Neubig. "Active retrieval augmented generation." In Proceedings of the 2023 Conference on Empirical Methods in Natural Language Processing, pp. 7969-7992. 2023.
>
> [3] Trivedi, Harsh, Niranjan Balasubramanian, Tushar Khot, and Ashish Sabharwal. "Interleaving retrieval with chain-of-thought reasoning for knowledge-intensive multi-step questions." In Proceedings of the 61st annual meeting of the association for computational linguistics (volume 1: long papers), pp. 10014-10037. 2023.
>
> [4] Jin, Bowen, Hansi Zeng, Zhenrui Yue, Jinsung Yoon, Sercan Arik, Dong Wang, Hamed Zamani, and Jiawei Han. "Search-r1: Training llms to reason and leverage search engines with reinforcement learning." arXiv preprint arXiv:2503.09516 (2025).

---

> > ### Comment · Reviewer_H7AX · 2025-11-27
> >
> > Thanks for your rebuttal. After reading your clarifications, I prefer to keep my rating.

---

> ### Author Response · Authors · 2025-11-27
> **Additional Evaluation Addressing Noise and Hallucination Concerns (W2, W3, W5)**
>
> We sincerely thank the reviewer for the careful re-assessment.
>
> *(We also note that in response to W4, we have **supplemented Point 5 in Rebuttal (1/2)** with results on T-REx and PopQA to further broaden the benchmark coverage.)*
>
>
> We understand that several of the key concerns involved the robustness of our KG-guided retrieval—specifically the **potential noise introduced by KG-Based Query Expansion (W2, W3)** and the need to evaluate **hallucination and truthfulness beyond token-level metrics (W3, W5)**.
>
> To strengthen the empirical evidence addressing these points, we have conducted an **additional evaluation using LLM-as-a-Judge protocol aligned with CRAG’s quality assessment framework [5]**,  which directly measures *Accuracy, Hallucination, Missing Information, and Truthfulness*.
>
> The results on 2WikiMQA and MuSiQue are shown below.
>
> **Table 2. LLM-as-a-judge Evaluation Results on 2WikiMQA and MuSiQue**
>
> (*Generator: Qwen2.5-7B-Instruct; Judge: GPT-4o-mini*).
>
> | Method | **2WikiMQA** |  |  |  | **MuSiQue** |  |  |  |
> |--------|--------------|----|----|-------------|--------------|----|----|-------------|
> |        | Accuracy | Hallucination | Missing | Truthfulness | Accuracy | Hallucination | Missing | Truthfulness |
> | No Retri          | 24.8 | 74.8 | 0.4 | -50.0 | 7.8  | 89.8 | 2.4 | -82.0 |
> | Vanilla RAG       | 33.2 | 49.6 | 17.2 | -16.4 | 9.6  | 78.6 | 11.8 | -69.0 |
> | KG-Infused RAG    | **45.6** | **47.4** | 7.0 | **-1.8**  | **16.0** | **72.8** | 11.2 | **-56.8** |
>
> **Addressing Weaknesses W2, W3, and W5**:
>
> - **Noise Control (W2, W3)**: The reduction in **Hallucination** across both datasets (e.g., 49.6% → **47.4%**, and 78.6% → **72.8%**) provides direct empirical evidence that our controlled spreading activation and triple filtering do **not** introduce noisy KG branches. Instead, the KG acts as a precision-oriented retrieval signal that improves grounding.
>
> - **Robustness and Truthfulness (W3, W5)**: The substantial improvement in **Truthfulness** (e.g., –16.4 → **–1.8**) shows that the performance gains extend beyond token-level overlap and correspond to more faithful, non-fabricated reasoning. This aligns with your request to evaluate model quality under more semantic and stringent criteria.
>
> Furthermore, because Truthfulness is defined as $\text{Truthfulness} = \text{Accuracy} − \text{Hallucination}$, the joint behavior of these two metrics is crucial.
>
> If the KG acted merely as an additional answer-bearing corpus, one would expect the well-known “knowledge inflation” pattern:
>
> - Accuracy ↑
> - Hallucination ↑ (due to broader but noisier textual coverage)
>
> However, our results show the **opposite** trend on both datasets:
> - Accuracy ↑, **and simultaneously**
> - **Hallucination ↓** .
>
> This behavior indicates that the KG is **not** functioning as an extra store of answer text, but instead as a **precision-oriented grounding signal** that improves retrieval relevance and reduces unsupported generations.
>
> We hope these semantic evaluations provide a clear and rigorous response to the concerns regarding noise, hallucination, and evaluation robustness.
>
> **If any additional questions arise, we would be glad to clarify them or provide further analysis.**
>
>
> [5] Yang, Xiao, Kai Sun, Hao Xin, Yushi Sun, Nikita Bhalla, Xiangsen Chen, Sajal Choudhary et al. "Crag-comprehensive rag benchmark." Advances in Neural Information Processing Systems 37 (2024): 10470-10490.

---

### Official Review · Reviewer_6B3T · 2025-10-30

**Soundness:** 2
**Presentation:** 3
**Contribution:** 2
**Rating:** 4
**Confidence:** 4

**Summary:**

This paper presents KG-Infused RAG, a framework that augments standard retrieval-augmented generation by integrating external knowledge graphs through a spreading-activation mechanism. It enriches query retrieval with structured entity relations from existing KGs (e.g., Wikidata5M) without requiring costly graph construction. Experiments on five QA benchmarks show consistent accuracy and efficiency improvements over baseline RAG and prior KG-based systems.

**Strengths:**

This paper addresses a key gap between corpus-based and KG-based RAG approaches, namely, how to combine structured and unstructured knowledge for retrieval and reasoning efficiently.

Demonstrates gains on multiple QA datasets and provides comparisons with both corpus-only and KG-constructed baselines.

The framework can potentially be applied to enhance other RAG systems.

**Weaknesses:**

1. The writing of the paper is not very clear: The spreading-activation process (e.g., weighting, termination, or entity-selection heuristics) is only partially described. The use of spreading activation is intuitively motivated but lacks rigorous analysis or ablation exploring why it outperforms simpler entity expansion methods.

2. All benchmarks are QA-centric; there is no evidence the approach generalizes to other tasks (e.g., reasoning, summarization, scientific retrieval). Also, ther paper does not compare with existing retrieval-based methods such as Search-O1, IRCOT, HippoRAG, etc.

3. The paper occasionally positions the framework as a general RAG paradigm, but the results are confined to standard QA datasets.

4. The proposed method is somehow straightforward and heuristic in nature. The technical depth is limited.

**Questions:**

## Minor suggestions:

Figures and formulas (e.g., Eq. 2–5) could be condensed; some prompts and hyperparameters are deferred to appendices without summary.

## Questions:

How are activation weights propagated through multi-hop entities—uniformly or via learned relevance scores?

How sensitive is performance to the choice of KG (e.g., Wikidata vs. domain-specific graphs)?

Could this approach generalize to non-QA tasks, or is it specialized to entity-centric reasoning?

---

> ### Author Response · Authors · 2025-11-25
> **Rebuttal to Reviewer 6B3T (1/2)**
>
> We thank the reviewer for the thoughtful feedback and for highlighting that our work “addresses a key gap” in combining structured and unstructured knowledge efficiently. We address the concerns below.
>
> # 1. Clarifying the Spreading Activation Mechanism
> We appreciate the opportunity to clarify this component and will improve Sec. 3.3 accordingly.
>
> **(1) Propagation / weighting**
>
> Our spreading-activation mechanism does not propagate numeric weights.
>
> Instead, the LLM performs semantic activation: at each round it selects triples relevant to the question, effectively functioning as a soft, semantic relevance estimator. This avoids the rigidity of graph-topology–based weighting while capturing query-dependent associations.
>
> **(2) Entity selection & termination**
>
> As stated in Sec. 3.3:
> - Activation stops when no new entities provide additional relevant information (i.e., $E_{q}^{i+1} = \emptyset$), or
> - when the predefined maximum number of rounds is reached.
>
> This prevents semantic drift and keeps computation bounded.
>
> **(3) Why multi-round activation outperforms 1-hop expansion**
>
> Thank you for raising this point. A single 1-hop expansion provides only the immediate neighbors of the seed entities, which limits the ability to surface semantically richer or multi-hop related concepts. This often results in entity sets that are broad but not sufficiently informative for multi-step reasoning.
>
> In contrast, multi-round activation enables the model to **progressively accumulate semantically relevant multi-hop cues**, guided by LLM-based triple selection at each step. This incremental process allows the model to retrieve facts that are more closely aligned with the underlying reasoning chains required by multi-hop questions, while naturally filtering out less relevant branches.
>
> To directly examine this, we conducted a **small diagnostic comparison** between 1-round and 2-round activation. The observed trend aligns with the above intuition: **2-round activation consistently provides more useful multi-hop signals**, improving both KG-guided query expansion (as reflected in Table 5 of the paper) and downstream knowledge augmentation.
>
> **Table 1. Performance of KG-Augmented Generation with 1 vs. 2 activation rounds (Qwen2.5-7B).**
>
> | Method | **2WikiMQA** |  |  | **MuSiQue** |  |  |
> |--------|--------------|----|----|------------|----|----|
> |        | Acc | F1 | EM | Acc | F1 | EM |
> | No-Retri               | 23.4 | 26.76 | 22.0 | 4.0 | 11.56 | 3.4 |
> | Vanilla RAG            | 30.6 | 33.54 | 27.2 | 5.2 | 11.49 | 3.6 |
> | KG-Aug Gen (1 round)   | 36.4 | 37.92 | 31.2 | 6.0 | 12.13 | 4.2 |
> | KG-Aug Gen (2 round)   | **41.8** | **42.37** | **33.6** | **8.0** | **14.0**  | **5.4** |
>
>
> # 2. Generalization beyond QA benchmarks
> Our main experiments rely on QA-style datasets, which aligns with common evaluation practices in RAG research (e.g., **Self-RAG, FLARE, IRCoT [1–3]**), where QA tasks capture retrieval, **reasoning, and evidence aggregation (summarization)**.
> To complement these results, we performed **small-scale** checks on:
> - **T-REx** (a slot filling task),
> - **PopQA** (a long-tail entity QA task)
>
> These datasets are widely used in prior RAG works such as Self-RAG and RAG-DDR [4].
>
> KG-Infused RAG shows consistent gains across these formats, suggesting the method is not confined to extractive multi-hop QA. We will summarize these findings briefly in the final version.
>
> **Table 2: Additional Evaluation on Complementary QA Tasks** (*Qwen2.5-7B-Instruct*).
>
> | Method | **T-REx (Slot Filling)** |  |  | **PopQA (long-tail entity QA)** |  |  |
> |--------|---------------------------|----|----|----------------------------------|----|----|
> |        | Acc | F1 | EM | Acc | F1 | EM |
> | No-Retri | 44.26 | 42.53 | 35.25 | 18.6 | 4.17 | 14.4 |
> | Vanilla RAG | 51.64 | 40.22 | 27.87 | 43.2 | 13.17 | 24.8 |
> | KG-Infused RAG | 51.64 | 40.17 | 28.69 | 62.4 | **20.99** | 41.0 |
> | KG-Infused RAG (DPO) | **56.56** | **44.88** | **33.61** | **62.8** | 19.63 | **41.2** |

---

> ### Author Response · Authors · 2025-11-25
> **Rebuttal to Reviewer 6B3T (2/2)**
>
> # 3. Technical depth and positioning
>
> **(1) Positioning relative to existing retrieval-based methods**
>
> We will clarify that Search-O1, IRCoT, and HippoRAG focus on improving corpus-based retrieval, while our goal is orthogonal:
> to make large pre-existing KGs usable by RAG pipelines without constructing new graphs.
>
> Thus, our contribution is complementary rather than overlapping.
>
> **(2) Technical contribution**
>
> Our method is deliberately lightweight, avoiding GNNs or costly KG construction (Table 22, page 22).
>
> Yet, through spreading activation, KG-guided query expansion, and KG-guided augmentation, it achieves consistent gains (3.9%–17.8% over vanilla RAG) and significantly faster inference than existing KG-based methods.
>
> This provides a practical **neuro-symbolic bridge** between curated KGs and RAG systems. We will emphasize this motivation more clearly.
>
> # 4. Minor suggestions
> Thank you for the helpful suggestions. In the final version, we will condense Equations (2–5) and provide a compact summary of prompts and hyperparameters to improve readability.
>
>
>
> [1] Asai, Akari, Zeqiu Wu, Yizhong Wang, Avirup Sil, and Hannaneh Hajishirzi. "Self-RAG: Learning to Retrieve, Generate, and Critique through Self-Reflection." In The Twelfth International Conference on Learning Representations.
>
> [2] Jiang, Zhengbao, Frank F. Xu, Luyu Gao, Zhiqing Sun, Qian Liu, Jane Dwivedi-Yu, Yiming Yang, Jamie Callan, and Graham Neubig. "Active retrieval augmented generation." In Proceedings of the 2023 Conference on Empirical Methods in Natural Language Processing, pp. 7969-7992. 2023.
>
> [3] Trivedi, Harsh, Niranjan Balasubramanian, Tushar Khot, and Ashish Sabharwal. "Interleaving retrieval with chain-of-thought reasoning for knowledge-intensive multi-step questions." In Proceedings of the 61st annual meeting of the association for computational linguistics (volume 1: long papers), pp. 10014-10037. 2023.
>
> [4] Li, Xinze, Sen Mei, Zhenghao Liu, Yukun Yan, Shuo Wang, Shi Yu, Zheni Zeng et al. "RAG-DDR: Optimizing Retrieval-Augmented Generation Using Differentiable Data Rewards." In The Thirteenth International Conference on Learning Representations.

---

> ### Author Response · Authors · 2025-11-27
> **Additional Semantic Evaluation: Supporting the Robustness of Our Integration Mechanism**
>
> To better understand why our integration mechanism and spreading-activation design lead to stronger downstream performance—and to offer a perspective that goes beyond token-level EM/F1—we additionally ran an evaluation using an **LLM-as-a-Judge** setup aligned with the CRAG quality framework [5].
>
> This evaluation reports Accuracy, Hallucination, and Truthfulness, providing a more semantic view of factual grounding and answer reliability.
>
>
> **Table 3. LLM-as-a-judge Evaluation Results on 2WikiMQA and MuSiQue** (*Generator: Qwen2.5-7B-Instruct; Judge: GPT-4o-mini*).
>
> | Method | **2WikiMQA** |   |  | **MuSiQue** |   |  |
> |--------|--------------|----|-------------|--------------|----|-------------|
> |        | Accuracy | Hallucination |  Truthfulness | Accuracy | Hallucination | Truthfulness |
> | Vanilla RAG       | 33.2 | 49.6  | -16.4 | 9.6  | 78.6  | -69.0 |
> | KG-Infused RAG    | **45.6** | **47.4**  | **-1.8**  | **16.0** | **72.8**  | **-56.8** |
>
>
>
> Since Truthfulness is defined as $\text{Truthfulness} = \text{Accuracy} − \text{Hallucination}$, the key observation is the **joint pattern**:
>
> KG-Infused RAG not only improves Accuracy **but also reduces Hallucination**.
>
> If the spreading-activation procedure simply injected more entities without meaningful structure, we would expect the opposite trend—namely, higher hallucination due to semantic drift. Instead, the reduction in hallucination appears consistently across datasets.
>
> This outcome suggests that our proposed components—LLM-guided triple selection, multi-round semantic activation, and KG-guided augmentation—contribute **genuine grounding signals** rather than heuristic noise. Together with the ablation studies in the main paper, these results further support the claim that spreading activation serves as an effective and robust integration strategy.
>
>
> [5] Yang, Xiao, Kai Sun, Hao Xin, Yushi Sun, Nikita Bhalla, Xiangsen Chen, Sajal Choudhary et al. "Crag-comprehensive rag benchmark." Advances in Neural Information Processing Systems 37 (2024): 10470-10490.

---

> > ### Author Response · Authors · 2025-12-04
> > **Generalization to Scientific Domains and Sensitivity to Domain-Specific KGs (W2, Q2)**
> >
> > To evaluate whether our method **generalizes to scienticfic tasks (W2) and domain-specific KG (Q2)** , we we conducted additional experiments in the biomedical/scientific domain using PrimeKG (biomedical-specific KG) [6] and PubMed (biomedical corpus). Results show that KG-Infused RAG continues to outperform Vanilla RAG:
> >
> >
> > **Table 4. Results on Biochemical domain tasks. (Generator: Llama3.1-70B-Instruct; KG: PrimeKG; Corpus: Pubmed).**
> >
> > | Method | **MMLU-Medical tasks** |
> > |--------|---------------------------|
> > |        | EM |
> > | No-Retri | 82.4 |
> > | Vanilla RAG | 83.2 |
> > | KG-Infused RAG | **85.4** |
> >
> > This demonstrates that our approach is **not tied to Wikipedia or Wikidata, and the integration mechanism transfers effectively to scientific retrieval and domain-specific knowledge graphs**.
> >
> >
> > [6] Chandak, Payal, Kexin Huang, and Marinka Zitnik. "Building a knowledge graph to enable precision medicine." Scientific Data 10, no. 1 (2023): 67.

---

### Official Review · Reviewer_ZcnM · 2025-11-01

**Soundness:** 2
**Presentation:** 3
**Contribution:** 2
**Rating:** 4
**Confidence:** 2

**Summary:**

The paper presents a framework designed to enhance RAG systems by integrating a large, pre-existing KG. The paper demonstrate promising experimental results across 5 QA benchmarks.

**Strengths:**

* The pipeline is clearly articulated and easy to follow.
* The proposed method is evaluated using a variety of LLMs.
* The paper provides comprehensive experimental results, including a clear ablation study that highlights the contributions of different model components.

**Weaknesses:**

* The novelty appears somewhat incremental, as previous research has also explored the use of KGs for QA and retrieval enhancement. The authors should more strongly emphasis what is new vs what is incremental. In particular, since the paper claims novelty in leveraging a pre-existing KG to enhance RAG, it should include comparisons with existing methods that also utilize pre-existing KGs (e.g., [1]). Additionally, it would be valuable to compare with the extensive body of work on QA systems that directly query existing KGs (e.g., [2]).
* All evaluation benchmarks are derived from Wikipedia, which overlaps significantly with Wikidata. As a result, the generalization and robustness of the proposed approach remain untested on more diverse datasets and realistic scenarios.
* The results for KG-based and non-KG-based methods are not directly comparable, as they use different corpora.

[1]. KERAG: Knowledge-Enhanced Retrieval-Augmented Generation for Advanced Question Answering.

[2]. QALD-10 — The 10th Challenge on Question Answering over Linked Data.

**Questions:**

none

---

> ### Author Response · Authors · 2025-11-25
> **Response to Reviewer ZcnM (1/2)**
>
> Thank you for the detailed and constructive feedback. We appreciate the reviewer’s positive assessment of our pipeline clarity, thorough experiments, and ablation design. We address the concerns below.
>
> # 1. Novelty of leveraging a pre-existing KG for RAG
> We agree that prior work has explored the use of KGs for QA. However, our contribution differs in both **task formulation** and **integration mechanism** from existing KGQA systems [1] and KG-enhanced retrieval works such as KERAG [2].
>
> **(1) Distinct task formulation from KGQA**
>
> KGQA systems directly answer questions using symbolic KG queries (e.g., SPARQL).
> In contrast, our framework is **corpus-centered RAG**, where KG signals:
> - guide multi-hop reasoning
> - refine query expansion (Sec. 3.4)
> - augment retrieved passages (Sec. 3.5)
>
> but **do not replace corpus grounding**.
> Thus, KGQA benchmarks such as QALD-10 evaluate a fundamentally different paradigm and are not suitable comparison points.
>
> **(2) Different from KERAG and other pre-existing KG methods**
>
> KERAG leverages a KG after converting questions into symbolic queries, and does not integrate KG signals into corpus-based retrieval steps.
> In contrast, our method provides a **0→1 integration capability**:
> > KG-Infused RAG is the first to integrate a pre-existing KG into corpus-based RAG through spreading activation, KG-guided query expansion, and KG-guided passage augmentation.
> This mechanism is orthogonal to both KGQA and KG query–based systems.
>
> # 2. Dataset overlap between Wikipedia and Wikidata
> We acknowledge the reviewer’s concern. However, we note the following.
>
> **(1) Overlap is expected and reflects common real-world settings**
>
> Many widely used KGs (Wikidata, DBpedia, Freebase) are constructed from Wikipedia.
> Most RAG baselines—including Self-RAG, FLARE, and IR-CoT [3–5]—also evaluate on Wikipedia-derived corpora for this reason.
>
> **(2) Our system does not answer directly from KG facts**
>
> KG facts are used only for:
> - **retrieval guidance via query expansion** (Sec. 3.4, Table 5)
> - **knowledge augmentation of passages** (Sec. 3.5, Table 4)
>
> The final answer is still generated **from corpus passages**. As shown in our ablation (Table 4), both KG-guided QE and KG-guided augmentation contribute independently, confirming that the improvements come from integration mechanisms rather than direct KG lookup.
>
> # 3. Comparisons between KG-based and non-KG-based methods
>
> We agree that adding an external KG expands the available information.
> However, our goal is not simply to add knowledge, but to answer the key research question:
> How can a large pre-existing KG be effectively integrated into corpus-based RAG?
> This integration is non-trivial: large KGs are dense and noisy, do not align structurally with corpus passages, and cannot be fed into RAG models directly.
> This is why prior works (GraphRAG, LightRAG, KG2RAG) rely on constructed small KGs rather than large external ones.
> Our contribution is an **integration pipeline** that includes:
> - controlled multi-hop spreading activation
> - KG-based query reformulation
> - KG-guided knowledge augmentation
>
> The ablation study (Table 4) shows that each component contributes significantly.
> Therefore, comparing to vanilla RAG is appropriate, as it isolates the effect of integration quality, not the amount of knowledge.
> We will clarify this motivation more explicitly in the revised paper.

---

> ### Author Response · Authors · 2025-11-25
> **Response to Reviewer ZcnM (2/2)**
>
> # 4. Additional Evaluation on Complementary QA Tasks
> To provide light additional evidence beyond the main evaluation domain, we performed **small-scale** tests on T-REx (slot filling) and PopQA (long-tail entity QA). Their task structures differ from the multi-hop QA benchmarks in the main paper and provide complementary validation conditions.
> KG-Infused RAG continues to show consistent gains:
>
> **Table 1. Additional Evaluation on Complementary QA Tasks** (*Qwen2.5-7B-Instruct*).
>
> | Method | **T-REx (Slot Filling)** |  |  | **PopQA (long-tail entity QA)** |  |  |
> |--------|---------------------------|----|----|----------------------------------|----|----|
> |        | Acc | F1 | EM | Acc | F1 | EM |
> | No-Retri | 44.26 | 42.53 | 35.25 | 18.6 | 4.17 | 14.4 |
> | Vanilla RAG | 51.64 | 40.22 | 27.87 | 43.2 | 13.17 | 24.8 |
> | KG-Infused RAG | 51.64 | 40.17 | 28.69 | 62.4 | **20.99** | 41.0 |
> | KG-Infused RAG (DPO) | **56.56** | **44.88** | **33.61** | **62.8** | 19.63 | **41.2** |
>
> # Summary
>
> We thank the reviewer again for the helpful comments.
> We will clarify the distinction from KGQA and KERAG, better explain the rationale for evaluating on Wikipedia-derived datasets, and emphasize that our contribution lies in the integration mechanism that makes large pre-existing KGs practically usable in corpus-based RAG.
> Additional complementary evaluations will be included in the final version.
>
> [1] QALD-10 — The 10th Challenge on Question Answering over Linked Data.
>
> [2] KERAG: Knowledge-Enhanced Retrieval-Augmented Generation for Advanced Question Answering.
>
> [3] Jiang, Zhengbao, Frank F. Xu, Luyu Gao, Zhiqing Sun, Qian Liu, Jane Dwivedi-Yu, Yiming Yang, Jamie Callan, and Graham Neubig. "Active retrieval augmented generation." In Proceedings of the 2023 Conference on Empirical Methods in Natural Language Processing, pp. 7969-7992. 2023.
>
> [4] Asai, Akari, Zeqiu Wu, Yizhong Wang, Avi Sil, and Hannaneh Hajishirzi. "Self-RAG: Learning to Retrieve, Generate, and Critique through Self-Reflection." In International Conference on Learning Representations. 2024.
>
> [5] Trivedi, Harsh, Niranjan Balasubramanian, Tushar Khot, and Ashish Sabharwal. "Interleaving retrieval with chain-of-thought reasoning for knowledge-intensive multi-step questions." In Proceedings of the 61st annual meeting of the association for computational linguistics (volume 1: long papers), pp. 10014-10037. 2023.

---

> ### Author Response · Authors · 2025-11-27
> **Additional Semantic Evaluation: Supporting Integration-Based Improvements (Addressing W3)**
>
> To further examine whether the performance gains of KG-Infused RAG arise from the **integration mechanism** rather than simply from having access to an additional knowledge source, we conducted a semantic evaluation using the **LLM-as-a-Judge protocol** aligned with the CRAG quality framework [6]. This evaluation assesses factual grounding and unsupported content, moving beyond token-level EM/F1 metrics.
>
>
> **Table 2. LLM-as-a-judge Evaluation Results on 2WikiMQA and MuSiQue** (*Generator: Qwen2.5-7B-Instruct; Judge: GPT-4o-mini*).
>
> | Method | **2WikiMQA** |   |  | **MuSiQue** |   |  |
> |--------|--------------|----|-------------|--------------|----|-------------|
> |        | Accuracy | Hallucination |  Truthfulness | Accuracy | Hallucination | Truthfulness |
> | Vanilla RAG       | 33.2 | 49.6  | -16.4 | 9.6  | 78.6  | -69.0 |
> | KG-Infused RAG    | **45.6** | **47.4**  | **-1.8**  | **16.0** | **72.8**  | **-56.8** |
>
>
> Because Truthfulness is defined as $\text{Truthfulness} = \text{Accuracy} − \text{Hallucination}$, the key empirical signal lies in the **joint behavior** of Accuracy and Hallucination.
>
> If the external KG were acting mainly as an additional **answer-bearing corpus**, one would expect the well-known “knowledge inflation” pattern:
>
> - Accuracy ↑
> - Hallucination ↑ (due to broader but noisier textual coverage)
>
> However, our results show the **opposite** trend on both datasets:
> - Accuracy ↑, **and simultaneously**
> - **Hallucination ↓** .
>
> This behavior indicates that the KG is **not** functioning as an extra store of answer text, but instead as a **precision-oriented grounding signal** that improves retrieval relevance and reduces unsupported generations.
>
> Therefore, the improvements stem from the **integration pipeline** rather than from corpus-size differences, directly addressing the reviewer’s concern in **W3**.
>
> [6] Yang, Xiao, Kai Sun, Hao Xin, Yushi Sun, Nikita Bhalla, Xiangsen Chen, Sajal Choudhary et al. "Crag-comprehensive rag benchmark." Advances in Neural Information Processing Systems 37 (2024): 10470-10490.

---

### Author Response · Authors · 2025-12-04
**Summary of Rebuttal and Discussions by Authors**

We sincerely thank all reviewers for their time and constructive feedback.
Below we summarize our rebuttal, to show how each weakness/question was addressed and what additional experimental evidence was provided.

---

## 1. Novelty and distinction from existing KG-based RAG / KGQA systems (ZcnM, H7AX)

Reviewers asked **whether our approach is sufficiently distinct from prior KGQA/KG-based RAG** (KERAG, Think-on-Graph).

We clarified that our method is not a KGQA or KG-only RAG method. It targets a fundamentally different objective:
- integrating pre-existing KGs into corpus-based RAG, without LLM-based KG construction.
- enabling corpus-based RAG methods to exploit external KGs
- leveraging the KG-facts via spreading activation mechanism, rather than performing symbolic querying.

**Our method is orthogonal to KGQA and to existing KG-constructed RAG methods** (eg., GraphRAG, LightRAG), and offering a capability (plug-and-play KG integration) that previous KG-based RAG do not provide.

---

## 2. Generalization across tasks and KGs (ZcnM, 6B3T, H7AX)

Reviewers **raised that our experiments were mainly on standard QA tasks and most benchmarks are derived from Wikipedia. They asked whether our method generalize across tasks and KGs**.

To address this, we added two sets of experiments:
- Complementary QA tasks: T-REx (Slot filling) and PopQA (long-tail entity QA). Results show consistent gains (T-REx: EM 27.87→33.61; PopQA: EM 24.8→41.2), which **indicates our method is not limited to standard multi-hop QA tasks**.
- Domain-specific tasks and KG: We included results on biomedical KG (PrimeKG) and datasets (MMLU medical subtasks). our method shows consistent gains (EM: 83.2→85.4), which indicates our method also **generalize to domain-specific KG and tasks**.

## 3. Hallucination, grounding, and whether gains come from “more knowledge” (ZcnM, W3; 6B3T, W1) and addressing hallucination concerns (H7AX, W2/W3/W5)

Reviewer **questioned whether the performance gains came merely from adding more knowledge** (from the KG)，and our evaluation **lacked hallucination-level analysis**.

To address this, we conducted an LLM-as-a-judge evaluation (CRAG-aligned). The results show that KG-Infused RAG simultaneously **increases Accuracy and reduces Hallucination**. This pattern is the opposite of the “knowledge inflation” effect (Accuracy↑ + Hallucination↑) ,
and therefore **confirm that the improvments arise from our controlled spreading activation and KG-guided mechanisms**, rather than simply adding more knowledge.

---

## 4. Explication of spreading-activation and noise control mechanism (6B3T, H7AX)
Reviewer 6B3T **asked a more detailed description of spreading activation and evidence that it outperforms simpler entity-expansion methods**, while H7AX raised **concerns that using KG subgraph for query expansion could introducd a lot of noise.

We clarified that the activation is semantic rather than numeric, where the LLM performs query-relevant triples selection at each round，and noise is controlled via semantic filtering, termination rule, and hyper-parameters of activation.

Empirically, ablations and more analyses (Tables 4–5 and 1-vs-2-round comparisons) show that KG-Based Query Expansion **improves retrieval performance (Recall@6) by 4–8.4%** across datasets, and KG-Augmented Generation further **strengthens answer accuracy**; removing either modules degrades performance notably.

These **findings suggest that our controlled spreading activation yield significant, low-noise KG signals** that outperform simpler expansion methods and **do not introduce much noise**.

---

## 5. Concerns about increased latency and computational expense (YuLU)
Reviewers **asked whether multi-round LLM calls increase inference cost and offset the savings from avoiding KG construction**.

We compared our inference cost with KG-Based RAG methods (as shown in Table 22, Appendix E):
- GraphRAG (global): ~284k tokens & 2.7s per query + hours of KG construction (even the corpus is small);
- KG-Infused RAG: ~3.9k tokens & 3.48s per query, with no KG construction cost.

Thus, our method not only avoids hours of KG construction but **also maintains low inference cost**.

---

## 6. Overall accessment

Across five main QA benchmarks, two additional datasets (T-REx, PopQA), domain-specific experiments (biomedical), semantic LLM-as-a-judge evaluations, ablations, and plug-in integrations with existing RAG systems, the evidence consistently shows that spreading-activation integration of a pre-existing KG into corpus-based RAG provides:
- significant and robust gains over Vanilla RAG (3.9–17.8%) and corpus-based RAG,
- better factual grounding with lower hallucination,
- more efficient than exising KG-based methods, and generalizability to existing RAG pipelines.

---

We carefully addressed each reviewer’s point and are confident that the main concerns have been adequately resolved.

---

### Meta-Review · Area_Chair_X6oV · 2026-01-05

**Summary:**

The main concern from the reviewers is the experiments of this paper. Specifically, the employed datasets are not fully convinced, and the compared methods are also questioned by one of the reviewer. Besides, one reviewer also mentioned that the contribution of this paper is somehow incremental.

**Reviewer Concerns:**

I think the examples of hallucination provided by the rebuttal are useful. But, the main concerns mentioned in the Summary Part still exist.

**Reviewer Scores:**

Due to the comprehensive rebuttal provided by the authors, I think some of the reviewers may slight increase their score. But overall speaking, this paper is below the acceptance bar of ICLR.

---

### Decision · Program_Chairs · 2026-01-26

Reject